# Preclinical validation of a live attenuated dermotropic Leishmania vaccine against vector transmitted fatal visceral leishmaniasis

Subir Karmakar[1], Nevien Ismail[1], Fabiano Oliveira [2], James Oristian[2], Wen Wei Zhang [3], Swarnendu Kaviraj[4], Kamaleshwar P. Singh[4], Abhishek Mondal[5], Sushmita Das[5], Krishna Pandey[5], Parna Bhattacharya[1], Greta Volpedo[6], Sreenivas Gannavaram [1], Monika Satoskar[1], Sanika Satoskar[1], Rajiv M. Sastry[1], Timur Oljuskin [1], Telly Sepahpour[1], Claudio Meneses[2], Shinjiro Hamano[7,8], Pradeep Das[5], Greg Matlashewski [3✉], Sanjay Singh[4✉], Shaden Kamhawi [2✉], Ranadhir Dey [1✉], Jesus G. Valenzuela [2✉], Abhay Satoskar[6✉] & Hira L. Nakhasi [1✉]

Visceral Leishmaniasis (VL), a potentially fatal disease is caused by *Leishmania donovani* parasites with no vaccine available. Here we produced a dermotropic live attenuated *centrin* gene deleted *Leishmania major* (*LmCen$^{-/-}$*) vaccine under Good Laboratory Practices and demonstrated that a single intradermal injection confers robust and durable protection against lethal VL transmitted naturally via bites of *L. donovani*-infected sand flies and prevents mortality. Surprisingly, immunogenicity characteristics of *LmCen$^{-/-}$* parasites revealed activation of common immune pathways like *L. major* wild type parasites. Spleen cells from *LmCen$^{-/-}$* immunized and *L. donovani* challenged hamsters produced significantly higher Th1-associated cytokines including IFN-γ, TNF-α, and reduced expression of the anti-inflammatory cytokines like IL-10, IL-21, compared to non-immunized challenged animals. PBMCs, isolated from healthy people from non-endemic region, upon *LmCen$^{-/-}$* infection also induced more IFN-γ compared to IL-10, consistent with our immunogenicity data in *LmCen$^{-/-}$* immunized hamsters. This study demonstrates that the *LmCen$^{-/-}$* parasites are safe and efficacious against VL and is a strong candidate vaccine to be tested in a human clinical trial.

[1] Division of Emerging and Transfusion Transmitted Diseases, CBER, FDA, Silver Spring, MD, USA. [2] Vector Molecular Biology Section, Laboratory of Malaria and Vector Research, National Institute of Allergy and Infectious Diseases, NIH, Rockville, MD, USA. [3] Department of Microbiology and Immunology, McGill University, Montreal, QC, Canada. [4] Gennova Biopharmaceuticals, Hinjawadi Phase II, Pune, Maharashtra, India. [5] Rajendra Memorial Research Institute of Medical Sciences, Patna, India. [6] Department of Pathology and Microbiology, Ohio State University, Columbus, OH, USA. [7] Department of Parasitology, Institute of Tropical Medicine (NEKKEN), Nagasaki University, Nagasaki, Japan. [8] The Joint Usage/Research Center on Tropical Disease, Institute of Tropical Medicine (NEKKEN), Nagasaki University, Nagasaki, Japan. ✉email: greg.matlashewski@mcgill.ca; Sanjay.Singh@gennova.co.in; skamhawi@niaid.nih.gov; Ranadhir.Dey@fda.hhs.gov; jvalenzuela@niaid.nih.gov; Abhay.Satoskar@osumc.edu; Hira.Nakhasi@fda.hhs.gov

Leishmaniasis is a complex disease transmitted by the bites of *Leishmania*-infected sand flies. Among the major medical manifestations of leishmaniasis, VL is the most severe form and is fatal if untreated[1]. Since curative drugs are toxic and often leads to drug resistance cases[2], a vaccine would be the best alternative. Patients who recover from leishmaniasis including VL develop protective immunity against reinfection, suggesting a road map for successful vaccine candidate where infection without pathology can induce a protective immunity[3,4]. However, currently there is no licensed human vaccine available for leishmaniasis.

The immunological protective mechanism in VL is complex. Cell mediated immunity is crucial for induction of host protective immune response, particularly Th1 immune response characterized by the production of IFN-γ and IL-12[5,6]. In contrast, disease progression is associated with a dominant Th2 type as well as IL-10 mediated response[7]. Although, this Th1/Th2 dichotomy is clear in cutaneous leishmaniasis, it is not well defined in visceral leishmaniasis[7,8]. A study of in vivo cytokine profile in VL patients showed elevated levels of IL-10 and IFN-γ expression, while the levels of IL-10 decreased markedly with resolution of disease[9,10].

Though several recombinant protein and DNA vaccines with or without adjuvant showed promising results in animal model, they failed to achieve satisfactory results in clinical trials[10–12]. Among the several vaccination strategies attempted against leishmaniasis, infection with low dose of live wild-type *Leishmania* promastigotes (leishmanization), was the only successful immunization strategy for CL in humans[13–15]. However, under the current regulatory environment such practice is not acceptable due to safety concerns[16,17]. Therefore, live attenuated *Leishmania* parasites that are nonpathogenic and provide a complete array of antigens of a wild-type parasite should induce the same protective immunity as "leishmanization" and thus would be an effective vaccine candidate.

Prior studies from our group demonstrated that laboratory grown live attenuated *centrin* gene-deleted *Leishmania donovani* parasite ($LdCen^{-/-}$) protects against *L. donovani* challenge in various preclinical animal models of VL[18–20]. Centrin a calcium binding protein is essential in the duplication of centrosomes in eukaryotes including *Leishmania*[21,22]. It is important to note that *centrin* gene deficient *Leishmania* parasites are attenuated only in intracellular amastigote stage and can be grown in promastigote culture[21].

Several studies supported by epidemiological evidence suggested that exposure to wild-type *L. major* parasites that causes localized self-resolving infection confer cross-protection against VL in various animal models and humans[23–25]. All these findings suggest that live attenuated dermotropic *Leishmania* parasites could be a promising vaccine candidate against VL and would be safer than using a viscerotropic live attenuated *L. donovani* strain. Using CRISPR gene editing, we have generated a marker-free *centrin* gene-deleted non-GLP-grade live attenuated *L. major* ($LmCen^{-/-}$) dermotropic parasite that is safe and protective when used as a vaccine in a mouse model for cutaneous leishmaniasis (CL)[26]. This represented a breakthrough because using CRISPR, it was possible to engineer $LmCen^{-/-}$ without any antibiotic resistance marker genes making it compliant for human vaccine trials.

Here, we investigated whether GLP-grade live attenuated dermotropic $LmCen^{-/-}$ parasite is an effective vaccine against potentially fatal VL transmitted by the bite of infected sand fly vector using preclinical hamster model. Hamster model is considered to be a gold standard of VL as they develop clinical symptoms like humans including succumbing to death[27]. Further, the long-lasting durability of protection of GLP-grade $LmCen^{-/-}$ was also tested against visceral infection. In addition, we identified potential novel $LmCen^{-/-}$ specific immune markers of protection which could be explored for immunogenicity in clinical trials. Finally, we tested immunogenicity of GLP-grade $LmCen^{-/-}$ parasites in the PBMCs isolated from healthy people living in VL non-endemic regions with possible implications for indicators of biomarkers in clinical trials.

## Results

**$LmCen^{-/-}$ immunization is safe in preclinical hamster model.** Prior to determining the protective efficacy of $LmCen^{-/-}$, we first evaluated the safety characteristics of $LmCen^{-/-}$ parasites in a hamster model. Hamsters were injected intradermally with either $LmCen^{-/-}$ promastigotes or wild-type *L. major* ($LmWT$) and monitored for lesion development up to 7 weeks (49 days) post injection and parasite loads were determined at the study end point by using serial dilution method. $LmCen^{-/-}$ injected hamsters did not develop any visible lesions up to 49 days of post injection (Fig. 1a, b). In contrast, hamsters injected with $LmWT$ parasites, developed ear lesions within 15 days of parasite injection that progressively increased in size (Fig. 1a, b). At 3 days post injection, the parasite load in the ears and dLNs was similar in both $LmWT$ and $LmCen^{-/-}$ injected hamsters (Fig. 1c, d). At 15 days post inoculation, we started to observe a significant difference in parasite load between these groups with $LmWT$ injected-hamsters displaying a higher parasite load compared to $LmCen^{-/-}$-injected hamsters (Fig. 1c, d). The difference in parasite load between these two groups was greater at days 28 and 49 with parasite numbers progressively increasing in $LmWT$ injected hamsters (Fig. 1c, d). Importantly, two of six $LmCen^{-/-}$ injected hamsters cleared the parasites by 28 days post injection from the ear. Furthermore, at day 49, no viable parasites were recovered from the ears or dLNs of hamsters injected with $LmCen^{-/-}$ parasites (Fig. 1c, d). However, when we measured the parasite burden by qPCR, low level of parasite DNA was observed both in the ear and dLN at 49-day post-$LmCen^{-/-}$ injection (Supplementary Fig. 1A–C). No viable parasites were recovered from the serial dilution of spleens and livers of either $LmCen^{-/-}$ or $LmWT$ injected hamsters at the timepoints tested.

To rule out the possibility that hamsters injected with $LmCen^{-/-}$ parasites may serve as reservoirs of these parasites to sand flies, we performed a xenodiagnosis test using noninfected sand flies at 2- and 8 weeks post injection (Fig. 1e), timepoints where parasites were present or undetectable (Fig. 1c, d), respectively, after injection with $LmWT$ or $LmCen^{-/-}$ parasites (Fig. 1f). Although 20% (2/10) of *L. longipalpis* sand flies fed on 2 weeks post $LmCen^{-/-}$ immunized hamsters were positive at 4 days of post-blood feeding but the parasites acquired did not survive as all flies (0/42) were negative at 8 days of post-blood feeding (Fig. 1g). Interestingly, none of the fed flies were positive for parasites at 4 days or 8 days post-blood feeding at 8 weeks post $LmCen^{-/-}$ immunization (Fig. 1g). In contrast, 50% (5/10) of sand flies exposed to 2 weeks of $LmWT$ post infected animals were *Leishmania* positive after 4 days post feeding and 25% (7/28) were positive after 8 days of post feeding. Sixty percent of sand flies were *Leishmania* positive after 4 days of post feeding and 30% were positive after 8 days of post feeding at 8 weeks of post infection. (Fig. 1g). Collectively, these results demonstrated that the attenuated $LmCen^{-/-}$ parasites are avirulent and thus safe and fail to establish an infection in sand fly vectors.

To assess the safety characteristics of $LmCen^{-/-}$ parasites, we investigated lesion development and survival of $LmCen^{-/-}$ parasites in immune-suppressed animals treated with dexamethasone (DXM) (Fig. 1h). Immune-suppressed hamsters previously injected with $LmCen^{-/-}$ parasites showed no lesions at the inoculation site (ear) compared to the ulcerative lesions that developed in $LmWT$ injected hamsters 4 weeks after immune-suppression (Fig. 1i, j). Only three of

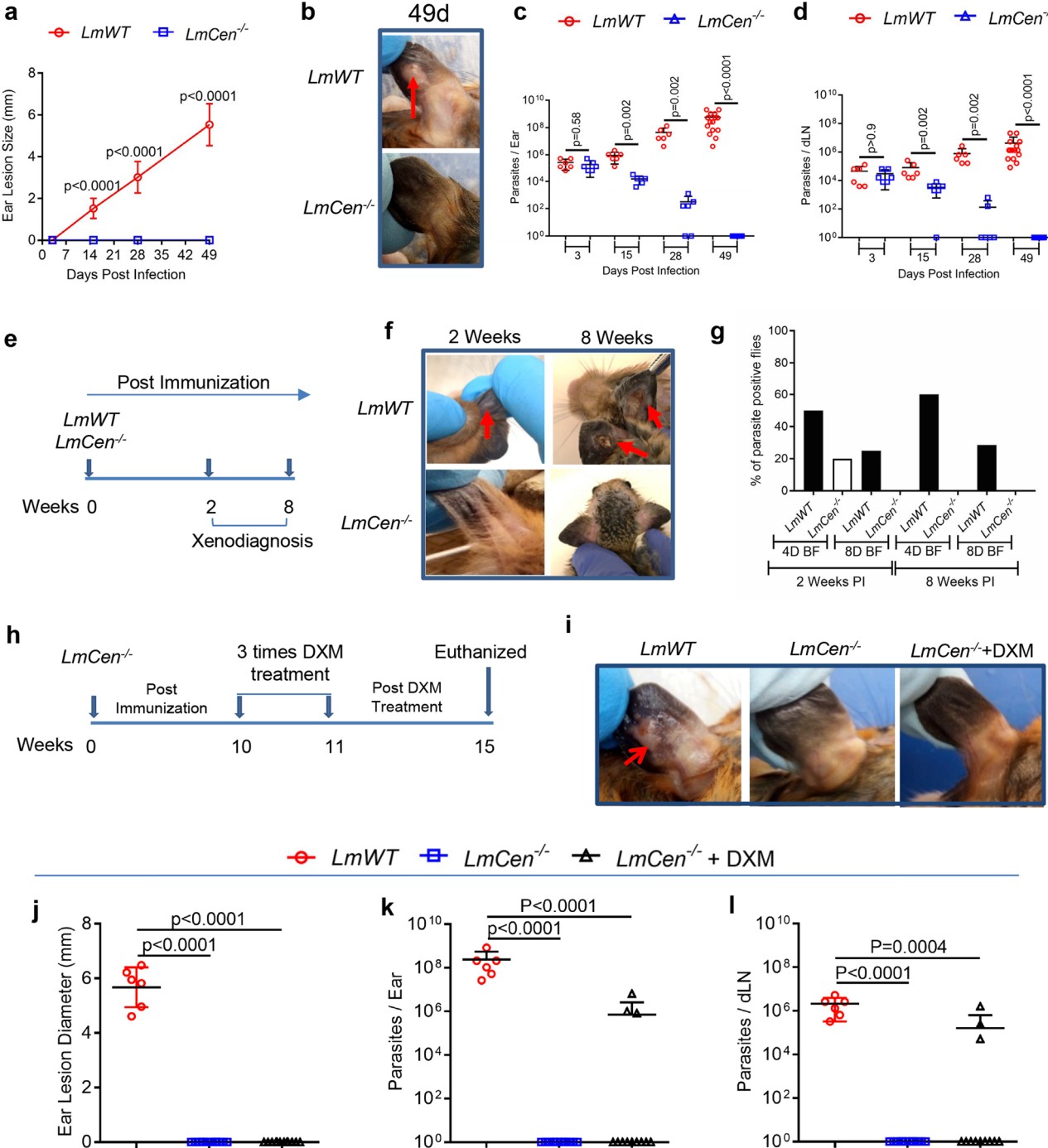

the 12 $LmCen^{-/-}$ injected/immune-suppressed hamsters had parasites in the inoculated ear (Fig. 1k) and in the dLN (Fig. 1l). As expected, all the $LmWT$-injected hamsters had significantly higher parasite loads in the ear (Fig. 1k) and in the dLN (Fig. 1l) compared to $LmCen^{-/-}$-injected animals (±immune-suppressed). To investigate whether $LmCen^{-/-}$ parasites isolated from immunosuppressed animals reverted to the wild-type genotype, we performed PCR analysis of the genomic DNA. We confirmed the absence of the *centrin* gene in the three isolates of $LmCen^{-/-}$ parasites recovered from immune-suppressed hamsters (Supplementary Fig. 2A, lanes, 3, 4, and 5, red arrow, similar to the original $LmCen^{-/-}$ parasites before immunization, lane 2 red arrow). We further examined whether the $LmCen^{-/-}$ parasites recovered from immune-suppressed hamsters had regained virulence by testing them in human monocyte-derived macrophages (hMDM). After 144 h infection, $LmCen^{-/-}$ parasites

were mostly cleared from the hMDM whereas the $LmWT$ parasites grew in hMDM (>6 parasites/hMDM Supplementary Fig. 2B, C). Collectively, these results demonstrate that the attenuated $LmCen^{-/-}$ parasites are safe, unable to revert to the wild-type form even in an immune-suppressed condition.

**$LmCen^{-/-}$ immunization induces protective immune response in preclinical hamster model.** Towards the analysis of systemic immune responses associated with the divergent phenotype observed in $LmWT$ and $LmCen^{-/-}$ infections characterized by the progressive nonhealing lesions and absence of lesions respectively, we analyzed the expression of several immune markers following stimulation with freeze–thaw *Leishmania* antigen (FTAg) of splenocytes. The expression profile of

**Fig. 1 Live attenuated *LmCen*$^{-/-}$ parasites do not cause any pathology in hamster model and do not cause any infection in sand flies. a** Lesion size was monitored every week in hamsters injected with $10^6$-total stationary phase either *LmWT* or *LmCen*$^{-/-}$ parasites by intradermal (ID) injection. Ear lesion diameters were measured at indicated days post inoculation. Results (SD) are representative cumulative effect of two (3 d, 15 d, and 28 d) to three (49 d) independent experiments, (*p* values were determined by Mann–Whitney two-tailed test). **b** Photographs of representative ears of *LmWT* and *LmCen*$^{-/-}$ immunized hamsters at 49 days post inoculation. Red arrow indicates the lesion development. **c, d** Parasite load in the ear (**c**) and draining lymph node (dLN) (**d**) of *LmWT* and *LmCen*$^{-/-}$ immunized hamsters were determined by serial dilution assay at 3, 15, and 28 days (*n* = 6/group of hamsters) and 49 days (*n* = 15/group of hamsters) post inoculation. Results (Mean ± SD) represent cumulative effect of two (15 d) to three (49 d) independent experiments (*p* values were determined by Mann–Whitney two-tailed test). **e** Schematic representation of xenodiagnoses to determine infectiousness of immunized hamsters for sand flies. **f** Photographs of representative ears of *LmWT* and *LmCen*$^{-/-}$ infected hamsters for xenodiagnoses at 2- and 8- weeks post inoculation. Red arrows indicate the lesion development. **g** After exposed with infected hamsters (*n* = 6/group), blood-fed flies were isolated, and parasite positive flies were identified after dissection of flies isolated from both *LmWT* and *LmCen*$^{-/-}$ infected groups at 4 days post-blood feeding and 8 days post-blood feeding. Results (Mean ± SD) are representative one experiment. **h** Schematic representation of immune-suppression by DXM treatment of *LmCen*$^{-/-}$ immunized hamsters. **i** Photographs of representative ears of *LmWT*, *LmCen*$^{-/-}$, and *LmCen*$^{-/-}$ + DXM treated hamsters. Red arrow indicates the lesion development. **j** Ear lesion diameters were measured after 4 weeks of DXM treatment (total 15 weeks post parasite infection) in *LmWT* (*n* = 6) and *LmCen*$^{-/-}$ (*n* = 12) and *LmCen*$^{-/-}$ + DXM (*n* = 12) treated hamsters. Results (Mean ± SD) represent cumulative effect of two independent experiments, 1 ear, total 6–12 hamsters per group (*p* values were determined by Mann–Whitney two-tailed test). **k, l** Parasite load in the inoculated ear (**k**) and dLN (**l**) of each group of hamsters (*LmWT*, *n* = 6; *LmCen*$^{-/-}$, *n* = 12; and *LmCen*$^{-/-}$ + DXM, *n* = 12) were determined by limiting dilution assay. Results (Mean ± SD) represent cumulative effect of two independent experiments (*p* values were determined by Mann–Whitney two-tailed test). BF Blood Fed, DXM Dexamethasone.

transcripts for pro-inflammatory Th1 type cytokines (IFN-γ, TNF-α, IL-1β, IL-12p40, and IL-6) transcription factors (T-bet and STAT1) and chemokines/their ligands (CXCL9) was significantly higher in the *LmCen*$^{-/-}$ immunized animals compared to *LmWT* injected hamsters (Fig. 2a, b; Supplementary Fig. 3). However, splenocytes from *LmWT* infected hamsters had a significantly higher expression of both anti-inflammatory (IL-4, IL-21, GATA3, and STAT6) as well as regulatory (IL-10 and Foxp3) transcripts compared to *LmCen*$^{-/-}$ immunized group (Fig. 2a, b; Supplementary Fig. 3). The ratio of IFN-γ to IL-10 was significantly higher in spleen of the *LmCen*$^{-/-}$ immunized group compared to *LmWT* injected animals (Fig. 2b). These results collectively suggest that *LmCen*$^{-/-}$ immunization induces a pro-inflammatory environment in the spleen of immunized animals. To further explore the immune networks activated as a result of the *LmCen*$^{-/-}$ infection compared to *LmWT* infection in a hamster, data from the 23 markers measured from these groups (Fig. 2b and Supplementary Fig. 4) were analyzed using Ingenuity Pathway analysis (IPA) utilizing the knowledge databases. IPA analysis showed that common canonical pathways were activated in both the infections dominated by Th1 pathways, DC maturation, IL-17 signaling (Supplementary Fig. 4A). These pathways are commonly reported in numerous studies with *L. major* parasites[28,29]. Interestingly, unlike *LmWT* parasites, *LmCen*$^{-/-}$ infection did not activate Th2 pathway consistent with our previous studies in mouse models[26] (Supplementary Fig. 4A, Supplementary Table 1). Further, the immune regulatory networks associated with the immune markers measured in our study showed distinct differences in the immunogenicity of *LmCen*$^{-/-}$ and *LmWT* infections as indicated by markers such as miR-130, PDK4, and CTSB predicted to be present only in the *LmWT* but not in the *LmCen*$^{-/-}$ infection and conversely ELF3 and ACKR2 genes in *LmCen*$^{-/-}$ infection only (Supplementary Fig. 4B, C). Analysis of the upstream regulators consistent with the observed expression pattern of the immune markers indicated a shared set of regulators between *LmWT* and *LmCen*$^{-/-}$ infections (Fig. 2c, Supplementary Table 2). However, *LmCen*$^{-/-}$ infection showed a distinct set of upstream regulators including several pattern recognition receptors such as STING, TLR6, CD14 and transcription factors such as IRF3, TRIM24 that were not observed in *L. major* infection (Fig. 2c). These upstream regulators are targets of future studies to elucidate the immune regulation associated with *LmCen*$^{-/-}$ parasites and may serve biomarkers of immunity in future clinical trials.

Furthermore, from the analysis of immunoglobulin subtypes one can predict the outcome of the immune response, a IgG1 dominant response predictive of Th2 and a IgG2a dominant response predictive for Th1 type of immune response[30]. We measured antileishmanial IgG1 and IgG2a from the serum of *LmCen*$^{-/-}$ or *LmWT* parasite infected hamsters at 49 days of post infection (Supplementary Fig. 5A, B). The level of IgG1 was significantly higher in *LmWT* injected hamsters compared to *LmCen*$^{-/-}$ immunized hamsters (Supplementary Fig. 5A). In contrast IgG2a was significantly higher in *LmCen*$^{-/-}$ immunized hamsters compared to *LmWT* infected hamsters (Supplementary Fig. 5B) resulting in higher ratio of IgG2a/IgG1 in *LmCen*$^{-/-}$ immunized hamsters (Supplementary Fig. 5C).

**_LmCen_$^{-/-}$-immunized hamsters induce a pro-inflammatory/Th1 type of immune response upon challenge with wild-type _L. donovani_.** Next, we wanted to investigate the efficacy of immunization with *LmCen*$^{-/-}$ parasites against visceral leishmaniasis induced by intradermal injection of *L. donovani* parasites in hamsters (Fig. 3a). The hamsters were immunized with *LmCen*$^{-/-}$ parasites and after 7 weeks of post immunization the animals were needle challenged with *LdWT* parasites and monitored for different periods (Fig. 3a). Analysis of the parasite load revealed significant control of parasite numbers in the spleen (Fig. 3b) and liver (Fig. 3c) of *LmCen*$^{-/-}$ immunized hamsters compared to nonimmunized-infected animals at all timepoints tested. Immunized hamsters showed ~1.5 log-fold reduced parasite burden in spleen and liver as early as 1.5-month post challenge. By 12 months post challenge, the parasite burden was reduced by ~5 log-fold for the spleen (Fig. 3b) and ~12 log-fold for the liver (Fig. 3c). Of note, at 12 months post challenge, 36% (4 of 11) of the spleens and 90% (10 of 11) of the livers from immunized animals had undetectable numbers of viable parasites. Taken together, these data demonstrate that *LmCen*$^{-/-}$ elicits protection against needle infection in a hamster model of VL.

To characterize the immune correlates of protection, we measured the gene expression profile ex vivo after antigen restimulation for the spleen by qPCR following 1.5 months after needle challenge with virulent *L. donovani* parasites. Evaluation of the immune response in the spleen (same genes that were tested before challenge) also showed a markedly increased expression of pro-inflammatory cytokine, chemokine, and transcriptional factor transcripts (IFN-γ, TNF-α, IL-12p40, T-bet, STAT1, and CXCR3) with a significant decrease in

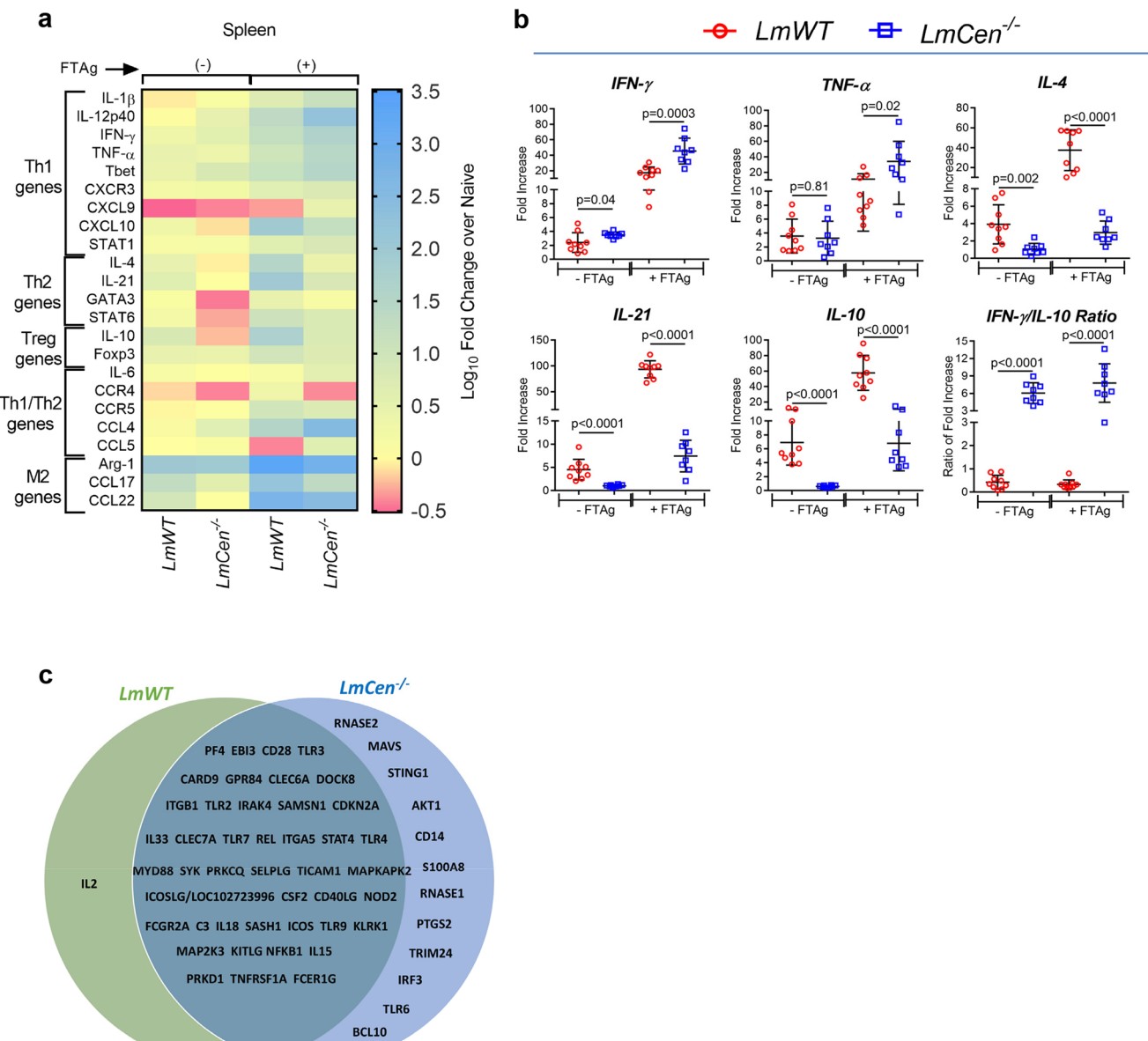

**Fig. 2 Immunogenicity of *LmCen*$^{-/-}$ parasites in hamsters. a** Heat map showing the differential gene expression in the spleen (with or without 24 h of *L. major* freeze–thaw antigen restimulation; ±FTAg) of *LmWT* infected and *LmCen*$^{-/-}$ immunized hamsters at 7 weeks post inoculation. Downregulation and upregulation of the transcripts are shown in blue, yellow and pink respectively. Transcripts are annotated in the left side according to their general functions. For heat map, results were shown as log$_{10}$ fold change over naive hamster. **b** Expression profile of IFN-γ, TNF-α, IL-4, IL-21, IL-10, and IFN-γ/IL-10 Ratio in the spleen which was evaluated by RT-PCR. The expression levels of genes of interest were determined by the $2^{-\Delta\Delta Ct}$ method; samples were normalized to either γ-actin expression or 18S RNA and determined relative to expression values from naive hamsters. The results were pooled of two independent experiments. Results (Mean ± SD) represent cumulative effect of two independent experiments (*n* = 4 for Naive, *n* = 9 for *LmWT*, and *n* = 8 for *LmCen*$^{-/-}$) (*p* values were determined by Mann–Whitney two-tailed test). **c** Ingenuity pathway analysis comparing the expression of the immune markers in *LmWT* and *LmCen*$^{-/-}$ groups showing upstream regulators as predicted by the IPA is shown. Most upstream regulators are shared between the *LmWT* and *LmCen*$^{-/-}$ infections as shown in the area of intersection. The statistical analysis is provided in Supplementary Information.

anti-inflammatory genes (IL-21 and STAT6) in the immunized-challenged group compared to the nonimmunized-challenged group (Fig. 3d, e; Supplementary Fig. 6). Of Note, expression of M2 macrophage phenotype genes (CCL17) and chemokine and chemokine receptor transcripts (CCR4 and CCL4) were higher in the nonimmunized-challenged group compared to nonimmunized-challenged group (Fig. 3d, Supplementary Fig. 6). Importantly, the ratio of IFN-γ to IL-10 was significantly higher in the spleen of the immunized-challenged group compared to nonimmunized-challenged animals (Fig. 3e). Collectively, these results indicate generation of a pro-inflammatory type of immune response following needle challenge in the *LmCen*$^{-/-}$ immunized animals that confers protection against virulent *L. donovani*.

**LmCen$^{-/-}$ immunization confers protection against sand fly-transmitted *L. donovani* challenge.** To determine the protective efficacy of *LmCen*$^{-/-}$ parasites against sand fly mediated VL infection, 7 weeks after a single intradermal immunization with *LmCen*$^{-/-}$ parasites, hamsters considered a gold standard challenge, were exposed to the bites of thirty *L. donovani*-infected sand flies in the contralateral ear. (Fig. 4a). The parasite load and intensity of the infection in the sand fly vector gut was assessed

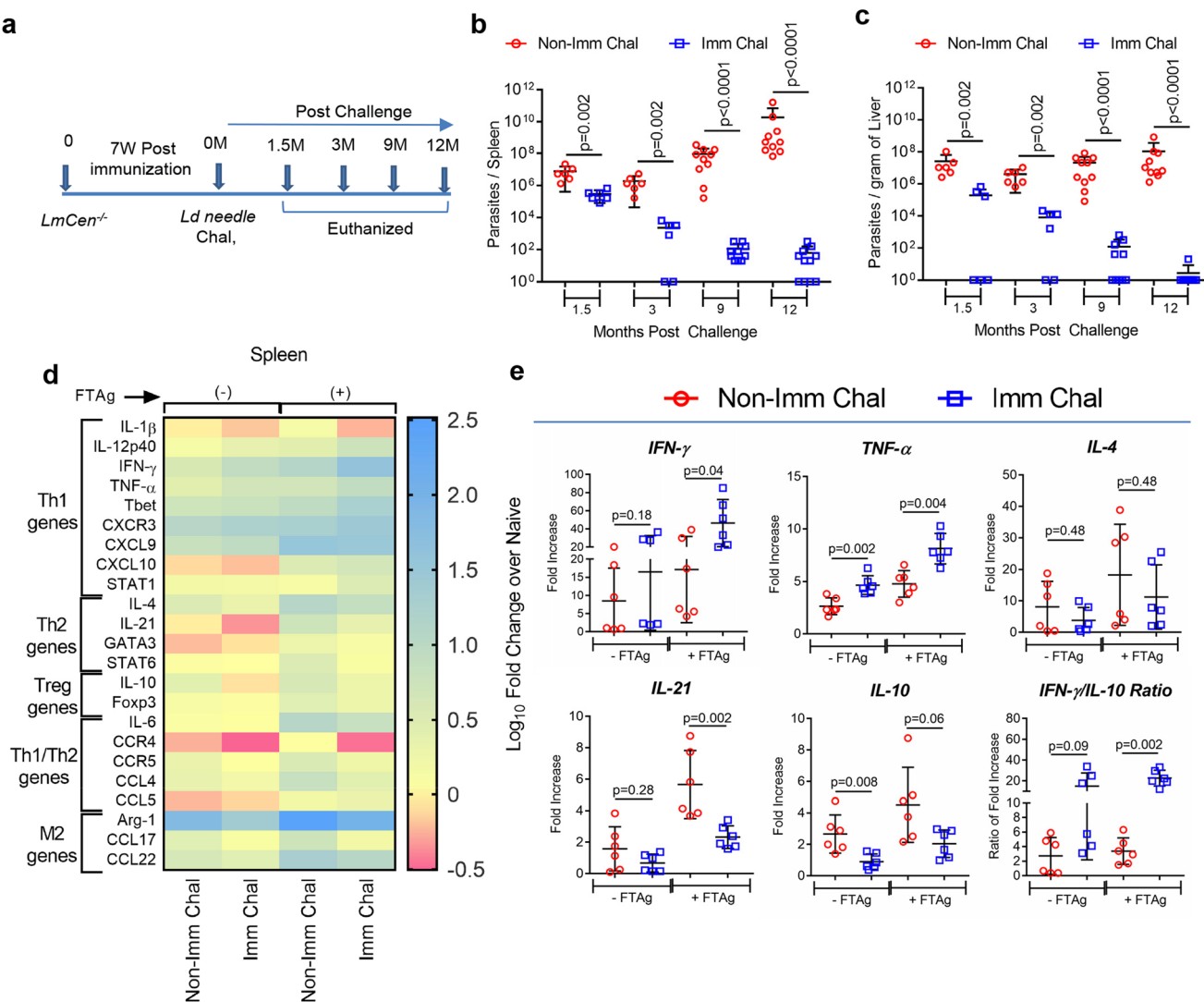

**Fig. 3 LmCen⁻/⁻ immunized hamster protects and induces pro-inflammatory type of immune response upon challenge with _L. donovani_. a** Schematic representation of experimental plan to determine the efficacy and immune response of _LmCen⁻/⁻_ parasites hamsters against _L. donovani_ via needle challenge. **b, c** Parasite load were determined by limiting dilution after various periods post challenge and expressed as number of parasites per Spleen (**b**) and per gram of Liver (**c**). _LmCen⁻/⁻_ immunized (Imm Chal) ($n = 6$ for 1.5 and 3 months and $n = 10$ for 9 and $n = 11$ for 12 months) and age-matched nonimmunized (Non-Imm Chal) hamsters ($n = 6$ for 1.5 and 3 months and $n = 10$ for 9 and 12 months) after various periods of post-needle challenge (1.5, 3, 9, and 12 month) with _L. donovani_. Results (Mean ± SD) are representative of cumulative effect of two independent experiments (_p_ values were determined by Mann–Whitney two-tailed test). **d** Heat map showing the differential gene expression in the splenocytes (with or without 24 h of _L. donovani_ freeze–thaw antigen restimulation; ±FTAg) of age-matched nonimmunized (Non-Imm Chal) and _LmCen⁻/⁻_ immunized (Imm Chal) hamsters after 1.5 months of _L. donovani_ challenge by needle injection ($n = 6$/group). According to the general function, transcripts are annotated in the left side. Downregulation and upregulation of the transcripts are shown in pink, yellow and blue, respectively. For heat map, results were shown as $\log_{10}$ fold change over naive hamster. **e** Expression profile of IFN-γ, TNF-α, IL-4, IL-21, IL-10, and IFN-γ/IL-10 Ratio in the spleen, which was evaluated by RT-PCR. The data were normalized to γ-Actin expression and shown as the fold-change relative to age-matched naive hamster. Results (Mean ± SD) represent cumulative effect of two independent experiments (_p_ values were determined by Mann–Whitney two-tailed test).

before transmission (Supplementary Fig. 7A). At 13 days after infection in the sand flies, the geometric mean parasite load per sand fly midgut was $>1.0 \times 10^5$ _L. donovani_ parasites, and the mean percent of metacyclic parasites per midgut was >80%, indicating a good quality infection in the sand fly (Supplementary Fig. 7A). The mean number of fed flies per group of immunized and nonimmunized hamsters were comparable (Supplementary Fig. 7B). Immunization of hamsters with _LmCen⁻/⁻_ protected animals against visceral infection as was demonstrated by lack of mortality and remained healthy up to 12–14 months post infection following sand fly challenge, the end of our study period, whereas all age-matched nonimmunized-challenged hamsters

had developed typical symptoms of VL following _L. donovani_ challenge within 6–14 month (Fig. 4b). Moreover, the progression of the infection was associated with a significant weight loss in all nonimmunized hamsters that developed severe VL with clinical symptoms compared to _LmCen⁻/⁻_ immunized hamsters beginning at 9 months post infection via sand fly challenge (Fig. 4c, Supplementary Fig. 7C). All the nonimmunized-challenged hamsters exhibited severe splenomegaly compared to _LmCen⁻/⁻_ immunized or naïve animals at 9–12 months post challenge (Fig. 4d left panel, Supplementary Fig. 7D). Correspondingly, numerous _L. donovani_ amastigotes were observed in the smears from spleen tissue of nonimmunized hamsters that

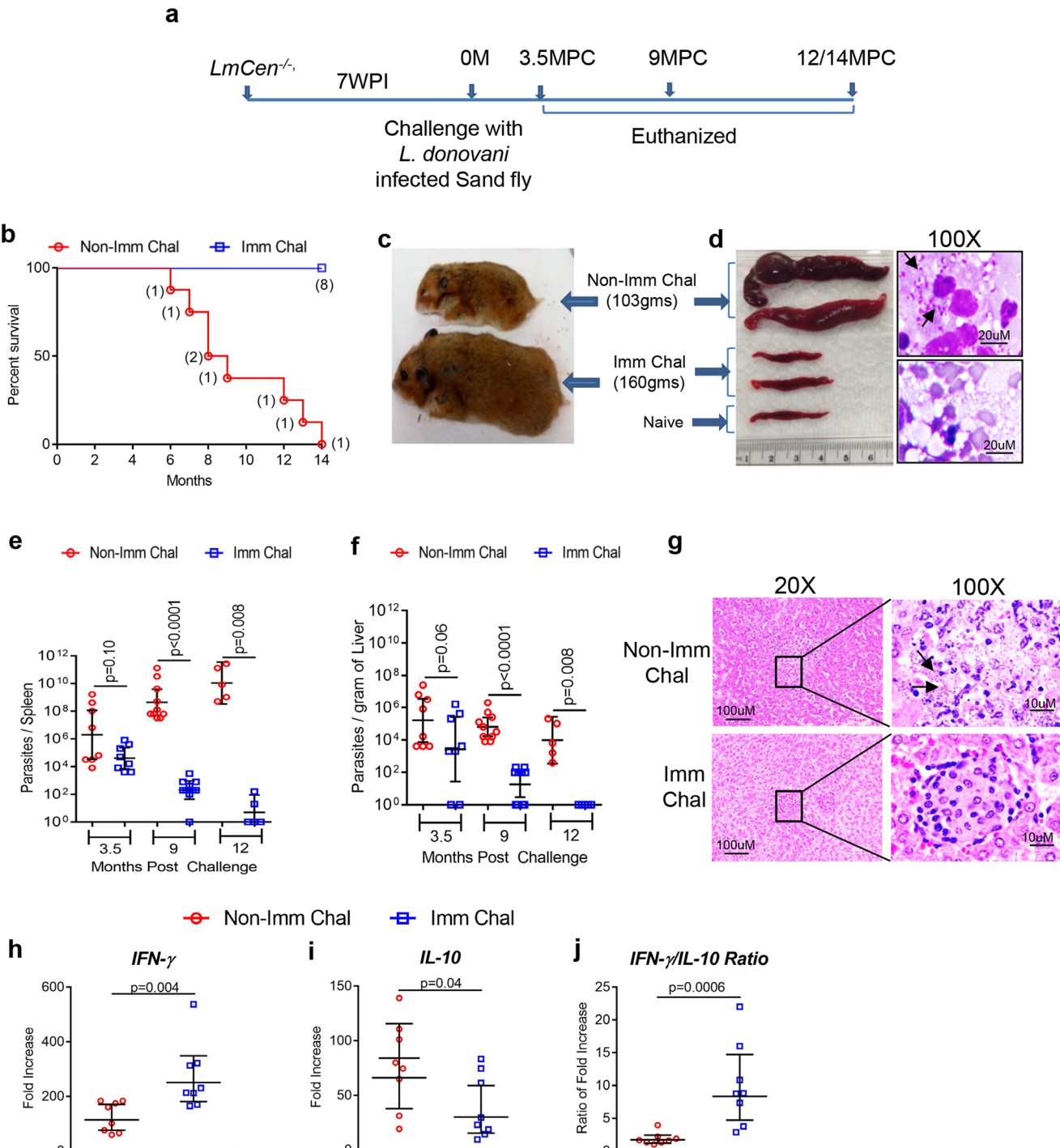

**Fig. 4 LmCen⁻/⁻ immunization confers protection against sand fly bite transmitted fatal L. donovani infection in hamsters. a** Schematic representation of the experimental plan. **b** Kaplan–Meier survival curves of *LmCen⁻/⁻*-immunized hamsters (Imm Chal; blue lines, *n* = 8) following challenge with *L. donovani*-infected sand flies and compared with age-matched nonimmunized-challenged group (Non Imm Chal; red lines, *n* = 8). **c** Picture of hamsters from each group after 9 months of sand fly-transmitted *L. donovani* challenge. **d** Photographs (left panel) of representative two spleen samples of both *LmCen⁻/⁻*-immunized and nonimmunized hamsters following 9 months post challenge as well as one age-matched naive hamster is shown. Right panel showing stamp smear of respective spleens stained with H&E, black arrows indicates intracellular parasites (bar-20 μm). **e, f** Parasite load in the spleen (**e**) and liver (**f**) (*n* = 8, *n* = 10, and *n* = 5 for 3.5M, 9M, and 12M post challenge, respectively, of nonimmunized and immunized-challenged groups). Results represent (the geometric means with 95% CI) cumulative effect of two (3.5MPC and 9MPC) and one (12MPC) independent experiment respectively (*p* values were determined by Mann–Whitney two-tailed test). **g** Liver sections are from animals at 3.5 months after challenge and stained with H&E (left panel ×20 and right panel ×100 magnification). **h–j** IFN-γ (**h**) and IL-10 (**i**) expression in the spleen of *LmCen⁻/⁻* immunized and age-matched nonimmunized hamsters (*n* = 8/per group) was evaluated by qPCR following 3.5 months post-*L. donovani*-infected sand fly challenge. The ratio of IFN-γ/IL-10 expression in the spleen was also determined (**j**). Results (the geometric means with 95% CI) represent the cumulative effect of two independent experiments (*p* values were determined by Mann–Whitney two-tailed test). WPI weeks postimmunization, MPC months post challenge.

were rare in comparison to the $LmCen^{-/-}$ immunized group (Fig. 4d, right panel). Furthermore, the parasite burden was controlled in both the spleen and the liver of $LmCen^{-/-}$ immunized hamsters compared to the progression of the infection in the nonimmunized group (Fig. 4e, f). At 12-month post challenge, a ~10-log-fold reduction in the parasite load in the spleen of $LmCen^{-/-}$ immunized hamsters compared to nonimmunized was observed (Fig. 4e). Notably, parasites were undetectable in the liver of immunized animals at 12 months post challenge (Fig. 4f). Since successful resistance to *L. donovani* in hepatic tissue is reflected by mature granuloma[31], we also examined liver sections at 3.5 months post challenge in both $LmCen^{-/-}$ immunized and nonimmunized hamsters. Histopathological analysis of the liver from nonimmunized animals revealed heavily parasitized Kupffer cells (black arrows) with few infiltrating lymphocytes resembling an immaturely formed granuloma (Fig. 4g, upper right panel). In contrast, $LmCen^{-/-}$ immunized hamsters exhibited well-formed granulomas comprised of concentric mononuclear cells (Fig. 4g, lower right panel).

To evaluate the immune response in $LmCen^{-/-}$ immunized hamsters following infected sand fly challenge, expression of two major cytokines, pro (IFN-γ), and anti-inflammatory (IL-10), was assessed in the spleen at 3.5-month post challenge. A protective Th1 immune response was observed as determined by a higher expression of transcripts of the pro-inflammatory cytokine IFN-γ (Fig. 4h) with concomitant lower expression of transcript of the anti-inflammatory cytokine IL-10 (Fig. 4i) from antigen restimulated spleen cells in the $LmCen^{-/-}$ immunized group compared to nonimmunized-challenged hamsters. This translated into a significantly higher ratio of IFN-γ to IL-10 in $LmCen^{-/-}$ immunized animals compared to nonimmunized animals (Fig. 4j) consistent with the immunological control of infection in cured VL patients[32,33]. Taken together, these data demonstrate that immunization with dermotropic $LmCen^{-/-}$ vaccine conferred protection against VL initiated via the natural mode of parasite transmission with infected sand flies by inducing a protective pro-inflammatory immune response.

**GLP-grade $LmCen^{-/-}$ parasites induce host protection in hamsters against wild-type *L. donovani*-infected sand fly challenge**. Advancing $LmCen^{-/-}$ parasites as a vaccine for potential human clinical trials will require parasites grown under current Good Manufacturing Practices (cGMP). Here, we produced GLP-grade $LmCen^{-/-}$ parasites in a bioreactor at a small industrial scale (1 L) with quality control characteristics (shown in Supplementary Table 3) and scalable for cGMP production for future clinical studies. Briefly, parasites were grown in 1 L bioreactor containing 0.5 L culture medium (shown in Supplementary Table 4), and vials (10 million cells/mL, in 1.8 mL volume) were prepared after 84 h of culture, when they reached a maximum cell density of 45–50 million cells per milliliter and stored in liquid nitrogen (Fig. 5a, b, c; Supplementary Table 5). The Cryo-preserved $LmCen^{-/-}$ parasites were thawed and used directly to vaccinate animals without further in vitro culture. Since laboratory-grade $LmCen^{-/-}$ protects against virulent *L. donovani* challenge, we first wanted to evaluate a comparative protective efficacy between laboratory-grade and GLP-grade $LmCen^{-/-}$ parasites against *L. donovani* challenge. Analysis of spleen and liver parasite loads after 9 months post needle challenge with *L. donovani* resulted in equivalent control of parasitemia in animals immunized with either GLP-grade or laboratory-grade parasites that was significantly reduced in comparison to nonimmunized control hamsters (Supplementary Fig. 8A, B). These data confirm that immunization with GLP-grade $LmCen^{-/-}$ parasites induce a similar protective immunity

as laboratory-grade $LmCen^{-/-}$ parasites against visceral infection in a preclinical animal model.

Next, we determined the protective efficacy of GLP-grade $LmCen^{-/-}$ parasites against gold standard sand fly challenge. Seven weeks post GLP-grade $LmCen^{-/-}$ immunized hamsters were exposed to *L. donovani*-infected sand flies bite in the contralateral ear. The parasite load and intensity of the infection in the sand fly vector gut was assessed before transmission as well as feeding score after transmission to hamsters (Supplementary Fig. 9A, B). GLP-grade $LmCen^{-/-}$ induced significant protection against sand fly challenge, as evidenced by a significant control of parasitemia both in the spleen and the liver 10 months post challenge. (Fig. 5e, f) and showed no splenomegaly (Fig. 5d). All GLP-grade $LmCen^{-/-}$ immunized hamsters survived the infected sand fly challenge and remained healthy up to 12 months post challenge, whereas two of five age-matched nonimmunized-challenged hamsters succumbed to *L. donovani* challenge within 12 month, the end of our study period (Fig. 5g) and rest of the animal showed signs of visceral leishmaniasis as indicated by splenomegaly (data not shown). Taken together these data validate that GLP-grade $LmCen^{-/-}$ parasites induce protective immunity and is consistent with the laboratory-grade $LmCen^{-/-}$ parasites against sand fly mediated visceral infection in a preclinical animal model.

**GLP-grade $LmCen^{-/-}$ immunization confers long-term protection against fatal *L. donovani* infection in hamsters**. To determine the durability of protection induced by GLP-grade $LmCen^{-/-}$, hamster were immunized with GLP-grade $LmCen^{-/-}$ parasites and challenged after 8 months post immunization with wild-type *L. donovani* through needle injection (Fig. 6a). Analysis of spleen and liver parasite loads after 8 months post-needle challenge showed a significant control of spleen and liver parasitemia. Parasite loads in immunized hamsters showed a ~3 log-fold and a ~2 log-fold reduction in the spleen (Fig. 6b) and liver (Fig. 6c), respectively, as compared to nonimmunized hamsters. These data confirm that a single intradermal immunization with GLP-grade $LmCen^{-/-}$ parasites induce a long-lasting protective immunity against visceral infection in a preclinical animal model.

**GLP-grade $LmCen^{-/-}$ parasites induce pro-inflammatory cytokines in human PBMCs**. We further evaluated the nature of the immune response induced by GLP-grade $LmCen^{-/-}$ in PBMCs of healthy individuals from the USA, non-endemic for VL (Fig. 7a). GLP-grade $LmCen^{-/-}$ parasites induced both IFN-γ (Fig. 7b) and IL-10 (Fig. 7c) in PBMCs of healthy individuals compared to their respective uninfected controls. However, the ratio of IFN-γ to IL-10 being significantly higher in the $LmCen^{-/-}$ group compared to the noninfected group (Fig. 7d), is consistent with the higher IFN-γ/IL-10 ratio observed in immunized hamster model (Fig. 2b)

## Discussion

There is an urgent need to have safe and efficacious vaccine for visceral leishmaniasis since drug treatment which are toxic and result invariably in developing parasite resistance. This is hampering the efforts to achieve *Leishmania* elimination goals set up by World Health Organization. It is known that patients who recover from leishmaniasis including VL develop protective immunity against reinfection, suggesting an alternative approach to drug treatment is to have a successful vaccine candidate where infection without pathology can induce a long-lasting protective immunity[3,4]. Previous studies demonstrated that infection with some *Leishmania* species can confer cross-protection against different species of *Leishmania*[20,23,24]. In addition,

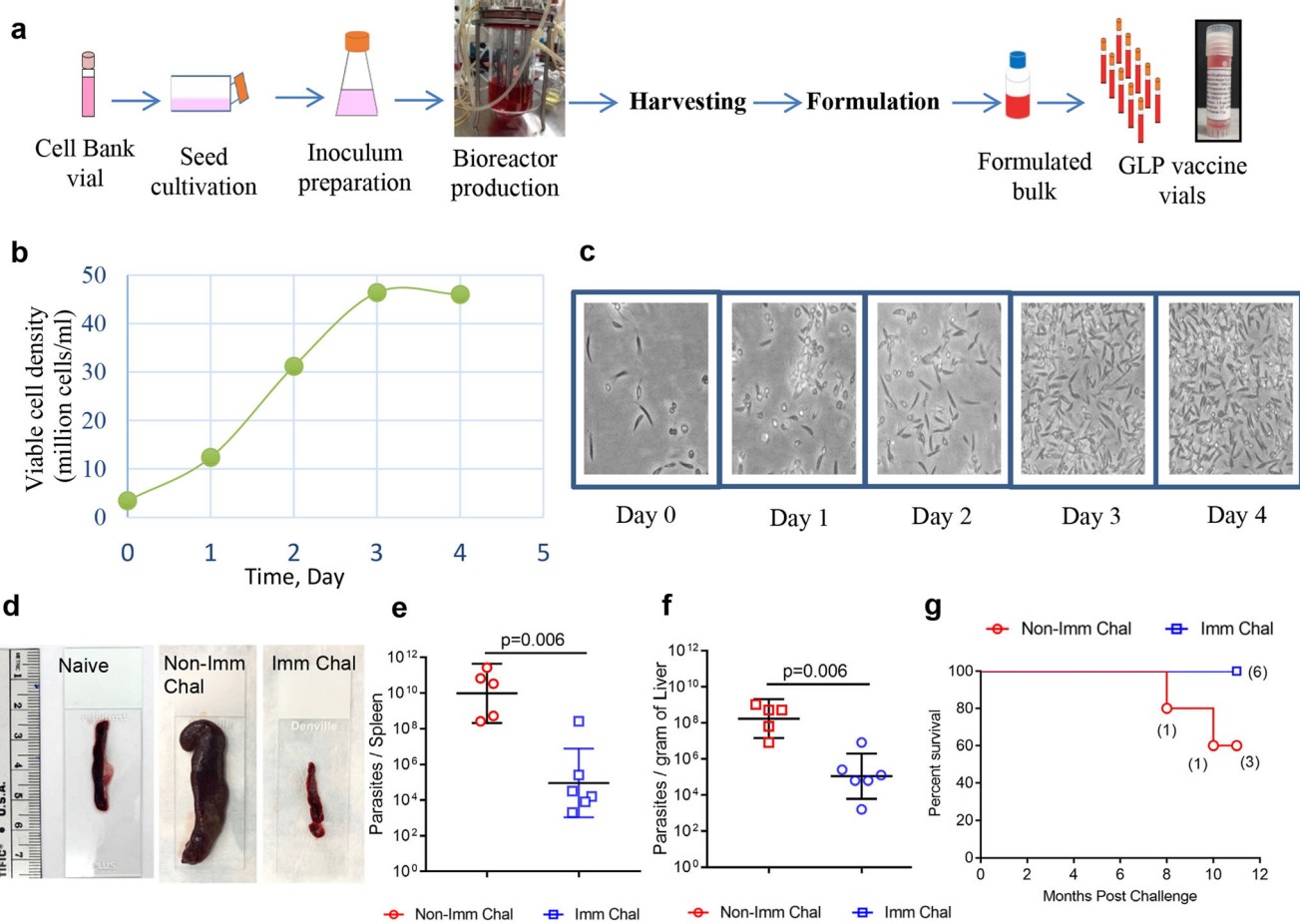

**Fig. 5 GLP-grade *LmCen*$^{-/-}$ parasites immunization confers protection in hamsters. a** Production of GLP-grade *LmCen*$^{-/-}$ parasite. **b** Growth behavior of parasite during bioreactor cultivation. **c** Morphology (magnification ×400) of the parasite during 4 days of bioreactor cultivation. **d** Photographs (left panel) of representative spleen samples of both *LmCen*$^{-/-}$-immunized and nonimmunized hamsters following 10 months post challenge as well as one age-matched naive hamster is shown. **e**, **f** Parasite load in the spleen (**e**) and liver (**f**) of hamsters either immunized with GLP-grade *LmCen*$^{-/-}$ (Imm Chal, n = 6) parasites or age-matched nonimmunized control (Non-Imm Chal, n = 5) were determined following 10 months of post challenge with *L. donovani*-infected sand flies. Results plotted as geometric means with 95% CI from one experiment (*p* values were determined by Mann–Whitney two-tailed test). **g** Kaplan–Meier survival curves of GLP-grade *LmCen*$^{-/-}$-immunized hamsters (Imm Chal; blue lines, n = 6) following challenge with *L. donovani*-infected sand flies and compared with age-matched nonimmunized-challenged group (Non-Imm Chal; red lines, n = 5).

epidemiological studies have suggested that exposure to wild-type dermotropic *L. major* parasites that causes localized self-resolving infection confer cross-protection against VL[23–25]. Taken together it suggests that there is a need to explore live attenuated dermotropic *Leishmania* parasites that are safe and do not have the potential of visceralization could be promising vaccine against VL. Previously, we have shown that laboratory grown live attenuated *centrin* gene-deleted dermotropic *Leishmania major* (*LmCen*$^{-/-}$) vaccine is safe and efficacious against virulent *L. major* infection[26].

In this study, we demonstrated that a single intradermal injection of a live attenuated *centrin* gene-deleted dermotropic *Leishmania major* (*LmCen*$^{-/-}$) vaccine either grown in the laboratory or produced under Good Laboratory Practices, confers robust and durable protection against lethal VL transmitted naturally via bites of *L. donovani*-infected sand flies and prevents mortality. We showed that *LmCen*$^{-/-}$ vaccine is safe in a hamster model, that prevents all the pathogenic features associated with clinical VL including hepato-splenomegaly and fatal outcomes[27]. Intradermal immunization with the live attenuated *LmCen*$^{-/-}$ did not develop any signs of disease pathology i.e., lesion development at the inoculation site compared to wild-type *L. major*

infection even in immune-suppressed animals. However, the persistence of a low number of *LmCen*$^{-/-}$ parasites, as observed in immune-suppressed immunized animals as well as detected by q-PCR in the nonimmune suppressed immunized animals, may be important for maintaining long-term protection as reported in other studies[34,35]. Experimental evidence suggests genetic exchange can happen between different *Leishmania* species in the sand fly vector[36,37], which may lead to the reversion of attenuated parasites to virulent form should such recombination occur in a sand fly and can be of safety concern for genetically modified live attenuated vaccines. To address that we showed that sand flies when fed on to *LmCen*$^{-/-}$ immunized animals failed to sustain the growth of parasites compared to WT infected animals in a xenodiagnoses experiment, suggesting possibility of genetic exchange of *LmCen*$^{-/-}$ parasites with the wild-type parasites remains an unlikely event in the *Leishmania* endemic areas. Thus *LmCen*$^{-/-}$ parasites are likely non-transmissible from immunized person to a naive nonimmune host.

Our result suggests that *LmCen*$^{-/-}$ immunization develops a pro-inflammatory environment in spleen with an abundant expression of Th1 cytokines. With *Leishmania* antigen-stimulation splenocytes from immunized animals significantly

up-regulate the expression of pro-inflammatory cytokine IL-1β, IFN-γ, along with TNF-α, and IL-12p40. In addition, a lower expression of anti-inflammatory cytokines IL-10, IL-4 and IL-21 in the vaccinated group further corelates with induction of host protective immunity and control of future infections[38,39]. The inter-relationship between transcription factors T-bet and STAT1, GATA3 and STAT6 and Foxp3, master regulators of Th1, Th2, and T-regulatory cell (Treg) development respectively, also determine the host immune response and outcome of the

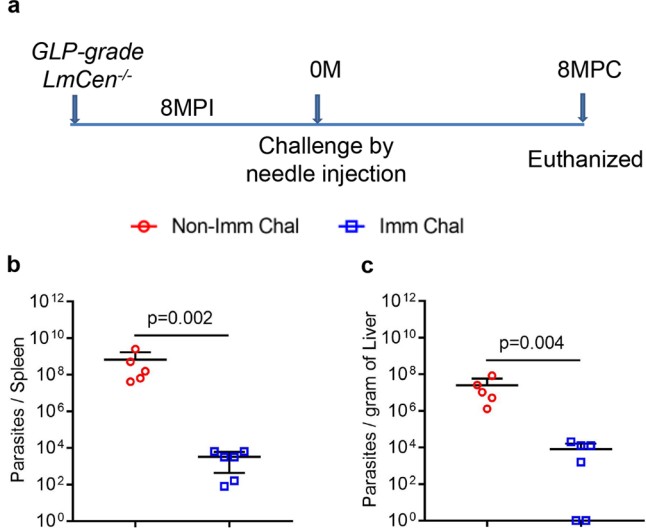

**Fig. 6 GLP-grade *LmCen⁻/⁻* immunization confers long-term protection against fatal *L. donovani* infection in hamsters. a** Schematic representation of experimental plan to determine the long-term protective efficacy of *LmCen⁻/⁻* parasites against *L. donovani* infection through needle injection. **b, c** Spleen (**b**) and liver (**c**) parasite burden of *GLP-grade LmCen⁻/⁻* immunized (Imm Chal, *n* = 6) and age-matched nonimmunized (Non-Imm Chal, *n* = 5) hamsters were determined at 8 months post-needle challenge. Results (means ± SD) are from one experiment (*p* values were determined by Mann–Whitney two-tailed test). MPI Months post immunization, MPC months post challenge.

disease[40,41]. Significant upregulation of T-bet, and STAT1 with concomitant downregulation of GATA3, STAT6, and Foxp3 in antigen restimulated spleen cells, suggests a biased Th1 phenotype in the vaccinated animals. Interleukin-6, a pleiotropic cytokine, showed elevated expression in antigen stimulated spleen cells, correlating with previous study demonstrating that IL-6 controls *Leishmania* disease progression by reducing the accumulation of regulatory T cells[42]. Further, the increased expression of IFN-γ, T-bet, STAT1, CCL5, and CXCL9 genes associated with M1 polarization of macrophages, along with the downregulation of Arg-1, CCL17 and CCL22 genes, a characteristic of M2 polarization of the macrophages[43], indicates that *LmCen⁻/⁻* immunization biases toward the M1 phenotype in vaccinated animals, and suggests that macrophages play an important role in the *LmCen⁻/⁻* mediated immune response to control parasitemia.

Analysis of the immunogenicity characteristics of *LmCen⁻/⁻* parasites revealed that overall common immune pathways are activated in *LmWT* and *LmCen⁻/⁻* parasites. However, the chemokine, cytokine panels tested in individual assays during post immunization reveal important differences associated with the non-virulent characteristics of *LmCen⁻/⁻* parasites especially with respect to Th2 pathway. However, analysis of the measurement of immune markers in the context of knowledge databases may potentially deepen our understanding of the immune response post immunization. Our analysis based on the human and mouse knowledge database in IPA restricted to immune cells showed that *LmCen⁻/⁻* infection induced distinct upstream regulators compared to *LmWT* infection. The broad range of upstream regulators predicted by the IPA including pattern recognition receptors and transcription factors observed in *LmCen⁻/⁻* infection suggests the strength of live attenuated parasites as potent immune-modulatory agents. The large number of upstream regulators shared with *LmWT* infection may point to the common immune regulation networks conserved in infections with either parasites suggest that immunization with *LmCen⁻/⁻* is analogous to leishmanization. These results highlight the distinct characteristics of live attenuated *LmCen⁻/⁻* parasites and their potential to generate a protective immune response compared to WT parasites which cause the disease.

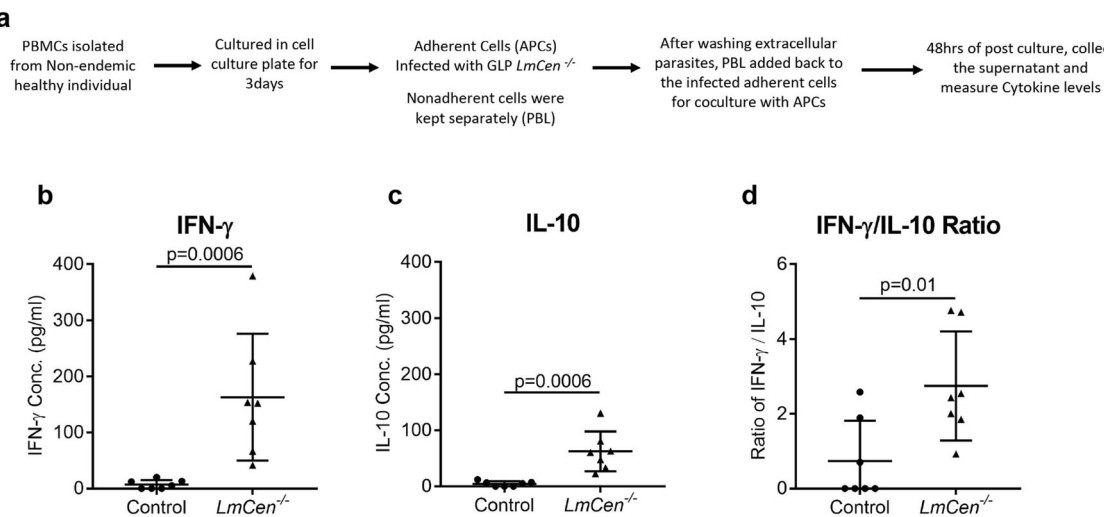

**Fig. 7 GLP-grade *LmCen⁻/⁻* induce host pro-inflammatory immune response in human PBMCs. a** Schematic representation of experiment plan to determine immune response of *LmCen⁻/⁻* in human PBMCs. **b–d** Scatter dot plots showing the levels (pg/ml) of pro-inflammatory (IFN-γ) (**b**) and anti-inflammatory (IL-10) (**c**) cytokine as well as IFN-γ/IL-10 ratio (**d**) in culture supernatant of PBMCs from Non-endemic region after 48 h of infection with *LmCen⁻/⁻* parasites by ELISA. Control group was left uninfected. Results (Mean ± SD) are representative of one experiment (*p* values were determined by Mann–Whitney two-tailed *t* test).

Similar analysis performed in other *Leishmania* vaccination regimens such as ChAd63-KH, a Chimpanzee Adenoviral vector expressing *Leishmania* vaccine antigens, revealed upstream regulators predominantly associated with interferon induced gene signatures such as IFNG, IRF7, IFNL1, IFNA2, STAT1, and IFNAR consistent with a viral vector induced response with IRF3 being the only common factor observed in *LmCen*[−/−] as well[10]. Interestingly, in both *LmWT* and *LmCen*[−/−] infections, IPA predicted activation of MAPK pathway which regulates expression of key immunoregulatory cytokines[44]. In contrast, similar analysis showed that MAPK activity was inhibited in ChAd63-KH vaccination suggesting important differences in the immune regulation associated with live attenuated parasites vaccines and viral vectored expression of parasite subunit vaccines[10]. While these remain predictions, the discovery of upstream regulators may allow us to identify novel immune mechanisms associated with *LmCen*[−/−] immunization in future studies. In addition, consistent with ours and other previous studies we also observed the pro-inflammatory environment in the spleen with a robust expression of Th1 cytokines in *LmCen*[−/−] infection including significantly up-regulated IFN-γ[5]. In contrast, IL-10 expression, a major regulatory cytokine which plays a crucial role in disease progression either by inhibiting the development of Th1 cells[45] or by blocking macrophage activation by IFN-γ[46], was significantly down-regulated in the vaccinated animals. In addition, a significantly higher ratio of IFN-γ to IL-10 was observed in the vaccinated animals than in the wild-type infected animals predictive of a vaccine efficacy[31].

We observed that *LmCen*[−/−] immunization induced significant host protection against *L. donovani* challenge either through needle injection or by infected sand fly challenge, as evidenced by the robust control of parasitemia both in the spleen and liver up to 12-month post challenge. This protection was associated to a pro-inflammatory or Th1 response observed systemically from antigen restimulated spleen cells after *L. donovani* infection initiated either by needle injection or sand fly bites. These results validated the efficacy of *LmCen*[−/−] as a vaccine candidate in contrast with other studies where protective efficacy of a vaccine against needle challenge is not always equally effective compared to sand fly vector mediated infection[47,48]. Importantly, *LmCen*[−/−] immunized animals exhibited a significantly higher ratio of IFN-γ to IL-10 as reported in cured VL patients as a potential regulatory mechanism to control visceral infections[32,33].

To explore whether *LmCen*[−/−] parasites can be used in future clinical trials as a vaccine candidate, we produced *LmCen*[−/−] parasites under GLP conditions, in a cGMP compliant facility. We demonstrated that immunization with GLP-grade *LmCen*[−/−] also results significant protection against *L. donovani*-infected sand fly challenge in a preclinical hamster model. In addition, GLP-grade *LmCen*[−/−] immunization also results in long-lasting protection against virulent *L. donovani* challenge, one of the essential characteristics of an efficacious vaccine. Moreover, GLP-grade *LmCen*[−/−] parasites induced both IFN-γ and IL-10 in PBMCs of healthy human subjects living in non-endemic regions. However, the higher IFN-γ/ IL-10 indicative of priming towards a Th1-biased immune response similar to our data in *LmCen*[−/−] immunized hamster studies prior to challenge. Corroborating our data, a recent study demonstrated that an antileishmanial DNA vaccine (ChAD63-KH) induced an IFN-γ-dominated immune response in healthy human volunteers[10]. Together, these data demonstrate the preclinical safety and protective efficacy of a GLP-grade *LmCen*[−/−] parasite vaccine that is ready to be tested in a first-in-humans clinical trial. The full spectrum of cytokine responses including their cellular sources and coordinated activities following immunization will be investigated in future clinical trials towards discovering biomarkers of vaccine efficacy.

In summary, the major advance reported within is that genetically modified marker-free GLP-grade dermotropic *Leishmania* vaccine is protective against a viscerotropic *Leishmania* validated in vivo against a natural vector challenge in a preclinical animal model of progressive VL and induces similar pro-inflammatory immune environment ex vivo in PBMCs of human subjects, indicative of protective response. Further, our immune analysis of *LmCen*[−/−] parasites based on the knowledge database (human and mice) in IPA restricted to immune cells showed that *LmCen*[−/−] immunization induced distinct upstream regulators compared to *LmWT* infection. The broad range of upstream regulators predicted by the IPA including pattern recognition receptors and transcription factors observed in *LmCen*[−/−] immunization suggests the strength of live attenuated parasites as a potent immune-modulatory agents and indicator of biomarkers in clinical trials. Taken together, this study demonstrates that *LmCen*[−/−], a safe vaccine[26] is efficacious against VL and should be advanced to first-in-human's clinical trials.

Finally, there are two perceived limitations to this study. One, although hamsters have been used as the gold standard animal model for VL, because of its similarity to humans in the outcome of mortality and morbidity, analysis of various parameters are constrained by the lack of availability of a broad range of immunological reagents for hamsters. Secondly, the immunological response of human samples from non-endemic region, presented in this study, is limited to PBMCs stimulated with GLP-grade material, and needs to be validated in clinical trials using GMP-grade material. We are currently in the process of manufacturing cGMP grade *LmCen*[−/−] parasites and are planning for a Phase 1 study in non-endemic regions and Phase 1 and Phase 2 clinical study of the *LmCen*[−/−] vaccine in endemic regions of VL to assess its safety and explore immunological correlates of protection. In future, we are also exploring to test the efficacy of *LmCen*[−/−] parasite vaccine in Controlled Human Infection Model (CHIM) that are currently under way.

## Methods

**Study design**. In this study we wanted to determine the vaccine efficacy of *centrin* gene-deleted *LmCen*[−/−] parasites against experimental VL in hamsters. All animal experimental procedures used in this study were reviewed and approved by the Animal Care and Use Committee of the Center for Biologics Evaluation and Research, U.S. Food and Drug Administration and the National Institute of Allergy and Infectious Diseases (NIAID) (http://grants.nih.gov/grants/olaw/references/phspolicylabanimals.pdf). The number of animals used in this study vary from 5 to 15 hamsters per group.

**Animals and parasites**. Six to eight-week-old female outbred Syrian golden hamsters (*Mesocricetus auratus*) were obtained from the Harlan Laboratories (Indianapolis, IN) and housed either at the Food and Drug Administration (FDA) animal facility, Silver Spring (MD) or National Institute of Allergy and Infectious Diseases (NIAID), Twin-brook campus animal facility, Rockville (MD) under pathogen-free conditions. The wild-type *L. donovani* (*LdWT*) (MHOM/SD/62/1S) parasites, wild-type *L. major* Friedlin (FV9) (*LmWT*) and *centrin* gene-deleted *LmCen*[−/−] (Friedlin strain) promastigotes were cultured as previously described[26,49].

**Hamster immunization and determination of parasite load**. Six to eight-week-old female hamsters were immunized with $10^6$ total stationary-phase either laboratory-grade or GLP-grade *LmCen*[−/−] parasites by intradermal injection in the ear in 10 μl PBS using a 29-gauge needle (BD Ultra-Fine). Control group of animals were infected with $10^6$ total stationary-phase *L. major* wildtype (*LmWT*) promastigotes. Lesion size was monitored every week by measuring the diameter of the ear lesion using a direct reading Vernier caliper. Parasite burden in the ear, draining lymph node (dLN) was estimated by limiting dilution analysis as described in previous studies[49].

**Xenodiagnosis**. Hamsters infected with *LmWT* and *LmCen*[−/−] parasites were xenodiagnosed at 2- and 8- weeks post infection. Briefly, 5 to 7-day-old unfed

female *Lutzomyia longipalpis* sand flies allowed to feed on the inoculated ear of anesthetized hamsters (the site of inoculation of *LmWT* and *LmCen*⁻/⁻ parasites) for 1 h in the dark. Blood-fed sand flies were separated and maintained in chambers under controlled conditions for 8 days. After 4 days or 8 days of exposure, midgut dissections were carried out, and examined by microscopy for the presence of live *Leishmania* parasites.

**Immunosuppression by dexamethasone treatment.** To determine the safety of *LmCen*⁻/⁻ parasites in immune-suppressive condition, 6- to 8-week-old hamsters were divided into three groups. Group-1 ($n = 6$) were infected with $10^6$ stationery phase *LmWT* parasites and Group-2 ($n = 12$) and Group-3 ($n = 12$) animals were immunized with $10^6$ stationery phase *LmCen*⁻/⁻ parasites in a 10 μl volume of PBS through intradermal routes (into the ear dermis). After 10 weeks of post infection, only Group-3 animals were treated with 2 mg/kg Dexamethasone sodium phosphate (Sigma-Aldrich) in PBS by subcutaneous injection three times, alternate days for 1 week. Four weeks after this treatment (total 15 weeks post infection), all the animals in three different groups were sacrificed and evaluated for parasite load by serial dilution as described[49]. Development of pathology and lesion size in the ear was assessed at 15 weeks post inoculation by measuring the diameter of the lesion.

Characterization of *centrin* gene-deleted parasites isolated from *LmCen*⁻/⁻ immunized and DXM treated group was done by Polymerase chain reaction. Total Genomic DNA was isolated from the parasites recovered from immune-suppressed hamsters as well as *LmWT* and *LmCen*⁻/⁻ parasites according to the manufacturer information (DNeasy Blood & Tissue kit, Qiagen). PCR was performed with *L. major centrin* gene specific primer (For-5′-ATGGCTGCGCTGACGGATGAAC AGATTCGC-3′; Rev-5′-CTTTCCACGCATCTGCAGCATCACGC-3′) which target the amplification of the 450-bp. A reaction mixture was prepared containing 10× Buffer (Invitrogen), 0.2 mmol/l each deoxyribonucleotide (Invitrogen), 1 μmol/l each primer, 1.25 U of Taq-polymerase (Invitrogen) and 200 ng of DNA samples in a final volume of 50 μl. The PCR conditions were as follows: denaturation at 94 °C for 3 min, followed by 35 cycles of 94 °C for 20 s, 58 °C for 20 s, and 68 °C for 35 s with a final extension of 68 °C for 5 min. The amplification reactions were analyzed by 1% agarose gel-electrophoresis, followed by ethidium bromide staining and visualization under UV light. DNA from the reference *plasmid* (PCR 2.1 TOPO) containing *centrin* gene was used as a positive control.

**Cytokine determination by real-time PCR.** At indicated timepoints, hamster spleens were collected, and single cell suspensions were made. Cells were stimulated with or without *L. major* or *L. donovani* Freeze thawed antigen (FTAg), and total RNA was extracted using PureLink RNA Mini kit (Ambion) 16 h of post stimulation. Aliquots (400 ng) of total RNA were reverse transcribed into cDNA by using random hexamers from a high-capacity cDNA reverse transcription kit (Applied Biosystems). Cytokine gene expression levels were determined by either TaqMan probe (TaqMan, Universal PCR Master Mix, Applied Biosystem) or SYBR green (Applied Biosystems) PCR using a CFX96 Touch Real-Time System (BioRad, Hercules, CA). The sequences of the primers (forward and reverse) and probes (5′ 6-FAM and 3′ TAMRA Quencher) used to detect the gene expression are shown in Supplementary Table 6[43,50]. The data were analyzed with CFX Manager Software. The expression levels of genes of interest were determined by the $2^{-\Delta\Delta Ct}$ method; samples were normalized to either γ-actin or 18S rRNA expression and determined relative to expression values from naive hamsters.

**Immunizations and needle challenge studies.** Six to eight weeks old female hamsters were immunized with $10^6$ stationary-phase either laboratory-grade or GLP-grade *LmCen*⁻/⁻ parasites in a volume of 10 μl PBS through intradermal routes. After 7 weeks of immunization, the animals were needle challenged (ID route) with virulent $5 \times 10^5$ metacyclic *Ld1S* promastigotes into the contralateral ear. As a control group, age-matched nonimmunized hamsters were similarly challenged with virulent $5 \times 10^5$ metacyclic *Ld1S* promastigotes. After various periods of post challenge (8- and 9-month post challenge), animals were sacrificed and organs (spleen and liver) were removed aseptically, weighed and parasite load was measured by limiting dilutions as previously described[49]. Hamsters were monitored daily during study, and their body weights were recorded weekly.

**Sand Fly challenge studies.** Female 2- to 4-day-old *Lutzomyia longipalpis* sand flies were infected by artificial chicken membrane feeding on defibrinated rabbit blood (Spring Valley Laboratories, Sykesville, MD) containing $5 \times 10^6$/ml *L. donovani* amastigotes, (freshly isolated from a sick hamster were used) supplemented with 30 μl penicillin/streptomycin (10,000 U penicillin/10 mg streptomycin) per ml of blood for 3 h in the dark as described previously[27]. Parasite loads and percentage of metacyclic per midgut were determined using hemocytometer counts. Sand flies on day 13- after infection were used for subsequent transmission to hamsters. Thirty flies with mature infections were applied to collateral ear of each *LmCen*⁻/⁻ immunized and age-matched nonimmunized hamsters through a meshed surface of vials held in place by custom-made clamps. During exposure to sand flies, hamsters were anesthetized intraperitoneally with ketamine (100 mg/kg) and xylazine (10 mg/kg). The flies allowed to feed for 2 h in the dark at 26 °C and 75% room humidity. As a qualitative measure of transmission, the number of blood-fed flies was determined. Each hamster received an average of 15-infected

bites per transmission. Hamsters were monitored daily during infection, and their body weights were recorded weekly. After various periods post challenge (3.5-, 9-, 10-, and 12 month), animals were sacrificed and organs (spleen and liver) were removed aseptically, weighed, and parasite load was determined by limiting dilutions method as previously described[49]. Furthermore, multiple impression smears of spleen tissue were prepared and examined for the presence of parasites in the tissue.

**Determination of parasite load by RT-PCR.** Seven weeks (or 49 days) post inoculation, DNA was purified from the ear and dLN of *LmCen*⁻/⁻ immunized hamster using DNeasy Blood & Tissue kits (Qiagen). The kinetoplast minicircle DNA (target DNA) of the parasite was amplified using seventy-five nanograms of sample DNA (template) using TaqMan probe-based RT-PCR (CFX96 Touch Real-Time System; BioRad, Hercules, CA) as described in previous studies[51].

**Manufacturing of GLP-grade *LmCen*⁻/⁻ vaccine.** In 1 L bioreactor, modified M199 medium supplemented with 10% FBS was used to produce GLP-grade *LmCen*⁻/⁻ parasites. To inoculate 1 L bioreactor, seed cultivation was started from cell bank. One cell bank vial was thawed and transferred entirely into T-25 cm² flask having 5 ml of medium. After 2–3 days, culture was expanded in T-75 cm² flask having 15 ml of medium. Further, after 2–3 days, culture was transferred into shake flask for inoculum development. After 2–3 days of shake flask cultivation, 1 L bioreactor was inoculated containing 500 ml medium. Bioreactor run was done for 4 days where culture reached its maximum cell density of $40–50 \times 10^6$ cells/ml. During bioreactor cultivation, parasites were found healthy and highly motile. At day 4, about 80–94 h when culture was in early stationary phase, parasites were harvested by centrifugation. Harvested parasites were washed with PBS and formulated to make vaccine. GLP vaccine vials were filled. One vaccine vial contained 1.8 ml of $10 \times 10^6$ cells/ml of live attenuated parasites.

**PBMC isolation and infection.** BMCs of seven different healthy donors from non-endemic regions were collected from ATCC, USA (PCS-800-011™). The viability of the PBMC was >98%, as checked by the trypan blue dye exclusion method. PBMCs were resuspended in complete RPMI-1640 medium (Invitrogen), supplemented with 100 U/ml penicillin (Invitrogen), 100 mg/ml streptomycin (Invitrogen), and 20 mM Hepes, pH 7.4 (Sigma-Aldrich), and plated at a density of $2 \times 10^6$ cells/ml in tissue culture dishes (Nalge Nunc Int., Rochester, NY, USA). These plates were incubated for 72 h for adherence at 37 °C in 5% $CO_2$. After this time, the autologous non-adherent cells were slowly aspirated out from the first plate and transferred to a second fresh plate and kept at 37 °C in 5% $CO_2$. Fresh complete RPMI-1640 medium was immediately added to the adherent cells in the first plate and kept at 37 °C in 5% CO2. The adherent cells were then either kept uninfected or were infected with *LmCen*⁻/⁻. After 6 h post infection, the unbound parasites were removed by washing twice with PBS. After washing, non-adherent autologous cells in the second plate were added back to the first plate and co-cultured with the uninfected/infected adherent cells for 48 h timepoints at 37 °C in 5% CO2. The supernatant was collected after the incubation, centrifuged at 400 g and the cell-free fraction was stored at −80 °C for cytokine ELISA.

**Cytokine ELISA.** The level of cytokines (IFN-γ and IL-10) in the cell-free culture supernatant were determined by using Human IFN-γ Uncoated ELISA kits (Cat No. 88-7316) and Human IL-10 Uncoated ELISA kits (Cat No. 88-7106) (Invitrogen, California, USA) respectively, as per manufacturers' protocol. Briefly, 96 welled ELISA plates (Corning Costar 9018, Corning, New York, United States) were coated with 100 μl/well of diluted capture antibody in coating buffer and was incubated overnight at 4°C. On the next day, the content of the wells was then aspirated out and washed 3 times with 1× Wash buffer (250 μl/well). Next, the wells were blocked with 200 μl of 1× ELISA diluent and incubated at room temperature. After an hour, the wells were washed twice with 1× Wash buffer.

Again, 100 μl of culture supernatant was added to each well and incubated overnight at 4 °C. On the following day, the contents of the wells were washed with 1X Wash buffer for 5 times. On completion, 100 μl of diluted detection antibody (pre-titrated, biotin conjugated anti-human IFN-γ, or IL-10 antibody) was added into each well and incubated for an hour followed by washing with 1X Wash buffer for five times. Thereafter, 100 μl of diluted Streptavidin-HRP (for IFN-γ) or Avidin-HRP (for IL-10) were then added in each well and incubated at room temperature for 30 min flowed by washing with 1× Wash buffer for seven times. After washing, 100 μl of 1× TMB substrate solution was added into each well and incubated at room temperature. After 15 min of incubation, the reaction was stopped by addition of 100 μl of Stop solution. The absorption was measured at 450 nm, and wavelength correction was performed at 570 nm. The concentrations of the cytokines were determined from respective linear standard curve and expressed as pg/ml.

**Antibody responses.** Specific antibody responses were measured by conventional enzyme-linked immuno-adsorbent assay (ELISA). Briefly, ELISA plates were coated overnight at room temperature with *Leishmania major* antigens (15 μg/ml). A serial dilution of the sera was carried out to determine the titer, which is defined as the inverse of the highest serum dilution factor giving an absorbance of >0.2.

The titers for the antibodies were determined using the following horseradish peroxidase-conjugated secondary antibodies: Rabbit anti-hamster IgG1-HRP, Rabbit anti-hamster IgG2a-HRP; Southern Biotech, Birmingham, AL; all with 1:1000 dilutions). SureBlue™ (KPL, Gaithersburg, MD) was used as a peroxidase substrate. After 15 min, the reaction was stopped by the addition of 100 µl of 1 M $H_2SO_4$, and the absorbance was read at 450 nm.

**Histological staining**. For histology, liver from hamsters were fixed in fixative solutions (10% buffered formalin phosphate solution). Sectioning and hematoxylin and eosin staining of all samples was done by Histoserv (Gaithersburg, MD). Stained sections were analyzed under the NIKON microscope. The Nikon-S-Element basic research software version 5.3.0 is used to acquire the images (Nikon Instruments Inc., Melville, NY).

**Ingenuity pathway analysis**. Two datasets of expression ratios and $p$ values for 23 immune markers including cytokines, chemokines and transcription factors from $LmCen^{-/-}$ ($n = 8$) and LmWT inoculated hamsters ($n = 9$) were uploaded to the IPA. All molecules were mapped by the software to its' database and expression values were automatically converted to fold-change values. A core expression analysis was performed on both datasets based on the fold-change values with fold-change cutoffs of $[-1,1]$ and a $p$ value cutoff of 0.05. The reference database selected was the Ingenuity Knowledge Base (Genes + Endogenous chemicals) and both direct and indirect relationships were considered. Default network settings with interaction networks box checked, include endogenous chemical box checked, molecules per network = 35, networks per network = 25, causal networks box checked, and score using causal paths only box checked. Node types and data sources were set to all and the confidence settings were set to experimentally observed only. Only human and mouse species were selected for both analyses. Only immune cells and immune cell lines were included in our analysis. The mutation settings were left at the default of all. Only entities that have a $-\log$ Benjamini–Hochberg $p$ value 1.3 (FDR of 0.05) are displayed. Only selected canonical pathways predicted to be activated are shown.

**Statistical analysis**. Statistical analysis of differences between groups was determined by unpaired two-tailed Mann–Whitney $t$ test, using Graph Pad Prism 7.0 software.

**Reporting summary**. Further information on research design is available in the Nature Research Reporting Summary linked to this article.

## Data availability
The data that support the findings of this study are available from the corresponding author upon reasonable request.

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

## Acknowledgements

Funding was provided from the Global Health Innovative Technology Fund, the Canadian Institutes of Health Research (to G.M.), intramural funding from CBER, FDA (to H.L.N.), and the Fonds de recherche du Québec—Santé (to P.L.). This research was supported, in part, by the Intramural Research Program of the NIH, National Institute of Allergy and Infectious Diseases (F.O., J.O., C.M, S.K., and J.G.V.). The findings of this study are an informal communication and represent the authors' own best judgments. These comments do not bind or obligate the Food and Drug Administration.

## Author contributions

S.Karmakar, N.I., F.O., J.O., W.W.Z., S.Kaviraj, K.S., A.M., S.D., K.P., P.B., G.V., S.G., M.S., S.S., R.S., T.O., T.S., C.M., and R.D. conducted experiments, analyzed data, and helped to write the paper. S.Karmakar, R.D., A.S., S.H., P.D., and S.Kamhawi, S.S., J.G.V., and H.L.N. designed experiments, analyzed data, and wrote the paper.

## Competing interests

The FDA is currently a co-owner of two US patents that claim attenuated *Leishmania* species with the Centrin gene deletion (US 7,887,812 and US 8,877,213). All other authors declare they have no competing interests.
