## [Peer Review File · Communications Biology]

Reviewers' Comments:

Reviewer #1:

Remarks to the Author:

These are very well done studies that represent an important advance in the development of a safe and effective vaccine against visceral leishmaniasis. Three critical aspects of the pre-clinical development were shown: 1) the live attenuated vaccine was shown to be effective against natural sand fly transmitted infection, 2) a hamster VL infection model was used, which permitted the efficacy to be demonstrated against the progressive, fatal form of disease, and 3) preliminary studies using GLP-grade parasites were presented. Additional studies looking at the immunogenicity of the vaccine in hamsters and in human PBMCs, add to the value of the studies.

Specific comments

In fig 1, the absence of any infected flies fed on the ears of hamsters infected with the LmCen^{-/-} parasites at the two week time point is a bit surprising since some of the ears contained over 10,000 parasites. A proper control would be to show that the LmCen^{-/-} parasites are able to survive and grow in flies. Do they establish infections following membrane feeds?

In fig 2, the rationale for the immunogenicity comparisons of the wild type and LmCen^{-/-} parasites based on the mRNA expression data is not so clear. The differences observed are not so easy to interpret because the concentrations of antigens to which the hamsters are exposed during infection are so vastly different. Furthermore, the biological significance of the specific differences observed is not clear. They would presumably be most relevant to how well the different exposures immunize hamsters against VL. Since hamsters experiencing cutaneous infections with the wild type strain were not evaluated, it is possible that their immunity against infected fly challenge is even better than the LmCen^{-/-}, in which case the responses that are unique to the wild type strain might be the more relevant.

Testing the efficacy of GLP-grade LmCen^{-/-} parasites is a valuable addition to the studies. There is some concern that in this experiment the spleen and liver parasite burdens at 10 months post-challenge, while significantly lower than the control hamsters, was still substantial in the vaccinated mice. These parasite numbers were over 100 times greater than the numbers shown in fig 3 at a similar time point in the hamsters immunized with the non-GLP-grade parasites. While these differences might be explained by differences in the infectious inoculum transmitted by the two populations of infected flies used in these experiments, the data do raise questions about the relative efficacy of the two vaccines, especially as only one time point was analyzed in hamsters immunized using the GLP-grade parasites, and the infections may be going up, not down at later time points.

Fig 4. The data sets referred to in the figure legend do not match the figure.

The primary response of the PBMCs from normal donors to stimulation using the LmCen^{-/-} parasites is informative but the interpretation of the data and the conclusion that the parasites are inducing a predominantly Th1 response with suppression of IL-10 is overstated. The relative concentrations of the two cytokines measured in the culture supernatants do not necessarily reflect the relative biologic activities of the cytokines – for example a lower concentration of IL-10 may be sufficient to initiate signaling in target cells compared to IFN γ . Since levels of IL-10 were in fact detected, there is no evidence that the IL-10 response is suppressed – compared to what? In addition, it is possible that stimulation of naïve human PBMCs *in vitro* with any *Leishmania* parasites, including *L. donovani* promastigotes, might induce the same primary response profile, so it is difficult to interpret what the signals mean in the context of vaccination.

Reviewer #2:

Remarks to the Author:

In this work, Karmakar et al. have analyzed the immunoprophylactic characteristics of a live vaccine based on the cutaneotropic species *Leishmania major*, attenuated by deletion of the genes

coding for Centrin. For this, two different hamster experimental models of VL have been used: the natural infection based on sand-flies and the infection using a needle. In anticipation of possible use in humans, the immunogenicity of the vaccine has been tested in vitro, using PBMC from healthy individuals.

This is a robust and well-planned work. The experiments are well done, and the number of animals employed is adequate. The greatest strength is the high degree of protection obtained in a natural model of infection that resembles the evolution of VL in humans.

There are some issues that deserve discussion, and some of conclusions should be reformulated.

Major points.

1.- Regarding biosecurity concerns.

It is not clear whether vaccination generates a persistent infection of the attenuated strain. The detection limit of the parasite load studies performed by limiting dilution should be indicated to avoid the apparent incongruity of saying in the results that there are no viable parasites at the site of infection (day 49 after vaccination; lines 147-152) and in the discussion that the infection is persistent (line 310-312) because of the presence of parasite DNA.

Does the infective challenge produce any effect on the parasite load at the vaccination site? In my opinion, all euthanized animals should be tested for the presence of vaccine parasites in different locations. Please note that concomitant immunity due to parasite persistence has been associated to the long-term immunity generated by leishmanization.

Another question. Can the presence of attenuated parasites in internal organs (liver, spleen, BM) be completely ruled out? DOI: 10.1016/j.actatropica.2005.09.007

Finally, If the infection is persistent, and thinking about a possible clinical trial in humans, shouldn't the effect of a possible immunosuppression on the parasite load of the vaccine parasites be studied?

These considerations should be resolved experimentally or if not possible at least be discussed in the text as limitations of the study.

2.- Regarding the parasite dependent cytokine response.

It would be interesting to know the production of IFN-gamma and IL-10 using splenocytes and the cells of the lymph nodes draining to the vaccinated ear after stimulation with Leishmania proteins (SLA) in the vaccinated hamsters.

Information about the SLA stimulation of spleen cells from challenged hamster (lines 219-229) are missed in the M/M section. Please, include these methods and indicate in the Fig.3 legend that cells were in vitro stimulated with SLA.

3.- Regarding the downregulation of the Th2 response.

The only evidence for the control of Th2 responses refers to the generic response pattern after immunization with the attenuated line in hamsters. In the rest of the experiments, the Th2 response has not been studied. Accordingly, the text must be corrected, especially in the abstract.

Respectfully, I do not agree with the interpretation of the results of the PBMC in vitro stimulation. The results show that after infection with the vaccine parasites, both IFN-gamma (F6b) and IL-10 (F6c) are produced. In other words, the vaccine generates the secretion of both cytokines. The next parts of the text must be corrected:

Results (277-280). GLP-grade LmCen-/- parasites induced higher (not lower) levels of IL-10 (Fig. 6c) in PBMCs of healthy individuals compared to their respective uninfected controls,

Discussion (369) GLP LmCen - / - do not induce IL-10 suppression in PBMCs, just the opposite occurs.

Consistent with the authors' conclusions, I agree that the production of IFN-gamma is greater than that of IL-10, but both cytokines are produced after parasite-driven stimulation

The confusion is even greater in the text of the abstract where it is stated that: "Downregulation of Th2 response in the PBMCs, isolated from healthy people from non-endemic region, upon LmCen - / - infection was also observed".

In addition to the already reported inaccurate interpretation of the results, the authors assign IL-10 the ability to define a Th2 response. Please note that this cytokine is produced by other cell types besides Th2 lymphocytes, even activated Th1 lymphocytes! Please edit the abstract to reflect the results more accurately.

To implicate down-regulation of the Th2 mediated responses in the induced protection in hamster or in PBMC assays it is necessary to include data regarding the expression of Th2 cytokines in the stimulation experiments discussed above (section 2) or in this section.

4.- Anti leishmania humoral responses.

In addition to my comments of section 3, it should be also considered that in the hamster model, as in humans affected by VL, the decrease in the antibody titer (IgG) against parasite proteins is correlated with the evolution of less severe forms of the disease as well as with vaccine-induced protection. Thus, it would be interesting to know the anti-leishmania antibody response elicited in the different groups of animals used in this work, or at least to know why the authors have not included these data.

Minor

The quality of figure 2 should be improved.

Consider editing lines 81-82 for more clarity: In contrast, disease progression is associated with a dominant Th2 type as well as IL-10 mediated responses.

Reviewer #3:

None

Reviewer #1 (Remarks to the Author):

These are very well done studies that represent an important advance in the development of a safe and effective vaccine against visceral leishmaniasis. Three critical aspects of the pre-clinical development were shown: 1) the live attenuated vaccine was shown to be effective against natural sand fly transmitted infection, 2) a hamster VL infection model was used, which permitted the efficacy to be demonstrated against the progressive, fatal form of disease, and 3) preliminary studies using GLP-grade parasites were presented. Additional studies looking at the immunogenicity of the vaccine in hamsters and in human PBMCs, add to the value of the studies.

Answer: We thank the reviewer for the appreciation of our work.

Specific comments

1. In fig 1, the absence of any infected flies fed on the ears of hamsters infected with the *LmCen*^{-/-} parasites at the two week time point is a bit surprising since some of the ears contained over 10,000 parasites. A proper control would be to show that the *LmCen*^{-/-} parasites are able to survive and grow in flies. Do they establish infections following membrane feeds?

*Answer: Sand flies do pick up the parasites from the hamster ears two weeks post-immunization with *LmCen*^{-/-} parasites from the inoculation site. Four days after exposure from a two-week *LmCen*^{-/-} immunized hamster ears, 20% sand flies were positive for the parasites. However, after eight days of exposure, out of 42 sandflies fed on the same hamster ears, none had any parasites. A revised New Fig. 1 g showing the 4 day post-inoculation data is added. The text has been revised accordingly (lines 162- 172).*

“Although 20% (2/10) of *L. longipalpis* sand flies fed on two weeks post *LmCen*^{-/-} immunized hamsters were positive at 4days of post blood feeding but the parasites acquired did not survive as all flies (0/42) were negative at 8days of post blood feeding (**Fig. 1g**). Interestingly, none of the fed flies were positive for parasites at 4days or 8days post-blood feeding at 8-weeks post *LmCen*^{-/-} immunization (**Fig. 1g**). In contrast, 50% (5/10) of sand flies exposed to 2-weeks of *LmWT* post infected animals were *Leishmania* positive after 4-days post-feeding and 25% (7/28) were positive after 8-days of post-feeding. Sixty % of sand flies were *Leishmania* positive after 4days of post-feeding and 30% were positive after 8-days of post-feeding at 8-weeks of post infection. (**Fig. 1g**). Collectively, these results demonstrated that the attenuated *LmCen*^{-/-} parasites are avirulent and thus safe and fail to establish an infection in sand fly vectors.”

Fig. 1

2. In fig 2, the rationale for the immunogenicity comparisons of the wild type and LmCen^{-/-} parasites based on the mRNA expression data is not so clear. The differences observed are not so easy to interpret because the concentrations of antigens to which the hamsters are exposed during infection are so vastly different. Furthermore, the biological significance of the specific differences observed is not clear. They would presumably be most relevant to how well the different exposures immunize hamsters against VL. Since hamsters experiencing cutaneous infections with the wild type strain were not evaluated, it is possible that their immunity against infected fly challenge is even better than the LmCen^{-/-}, in which case the responses that are unique to the wild type strain might be the more relevant.

Answer: *The rationale for the comparative analysis of immunogenicity between LmWT and LmCen^{-/-} parasites was to show the difference in the phenotypes of the LmWT parasites that cause the disease versus LmCen^{-/-} parasites that do not cause the disease (Fig 1). We have now revised the text (lines 193-207) on the analysis of the differences in the immune responses between LmWT and LmCen^{-/-} parasite infections by including the underlying data in New Figures 2a-b and supplementary Figure 3.*

“LmCen^{-/-} immunization induces protective immune response in preclinical hamster model : Towards the analysis of systemic immune responses associated with the divergent phenotype observed in LmWT and LmCen^{-/-} infections characterized by the progressive non-healing lesions and absence of lesions respectively, we analyzed the expression of several immune markers following stimulation with freeze-thaw *Leishmania* antigen (FTAg) of splenocytes. The expression profile of transcripts for pro-inflammatory Th1 type cytokines (IFN- γ , TNF- α , IL-1 β , IL-12p40 and IL-6) transcription factors (T-bet and STAT1) and chemokines/their ligands (CXCL9) was significantly higher in the LmCen^{-/-} immunized animals compared to LmWT injected hamsters (**Fig. 2 a, b**; Supplementary **Fig. 3**). However, splenocytes from LmWT infected hamsters had a significantly higher expression of both anti-inflammatory (IL-4, IL-21, GATA3 and STAT6) as well as regulatory (IL-10 and Foxp3) transcripts compared to LmCen^{-/-} immunized group (**Fig. 2 a, b**; Supplementary **Fig. 3**). The ratio of IFN- γ to IL-10 was significantly higher in both the spleen of the LmCen^{-/-} immunized group compared to LmWT injected animals (**Fig. 2b**). These results collectively suggest that LmCen^{-/-} immunization induces a pro-inflammatory environment in the spleen of immunized animals.”

Based on the RT-PCR data in New Figures 2a-b and supplementary fig 3, we have performed IPA analysis (supplementary fig 4 a-c) and represented the results in Figure 2c highlighting the upstream regulators predicted in LmCen^{-/-} infection compared to LmWT infection. Revised text lines 193-207.

The observed differences in the immune response are likely not due to the differences in antigenic load between LmWT and LmCen^{-/-} infections. If it were the case, all the markers would have shown a strong enrichment in LmWT infection compared to LmCen^{-/-} infection. Several markers such as IFN- γ and TNF are highly enriched in LmCen^{-/-} despite the differences in parasite load compared to LmWT infection.

Similar differences in the immune response have been reported in the literature that have attributed to the nature of the vaccine antigens compared to wild type infection that result in distinct outcomes.

Unlike C57Bl/6 mouse model which is resistant to progressive disease (Zhang et al Nat Comm, 2020 10:3461.), LmWT infection in a hamster model causes progressive non-healing disease as shown in Fig 1. B. LmWT infection thus cannot be used as a vaccine against VL challenge and protection results cannot be compared to LmCen^{-/-} immunization.

Fig. 2

3. Testing the efficacy of GLP-grade *LmCen*^{-/-} parasites is a valuable addition to the studies. There is some concern that in this experiment the spleen and liver parasite burdens at 10 months post-challenge, while significantly lower than the control hamsters, was still substantial in the vaccinated mice. These parasite

numbers were over 100 times greater than the numbers shown in fig 3 at a similar time point in the hamsters immunized with the non-GLP-grade parasites. While these differences might be explained by differences in the infectious inoculum transmitted by the two populations of infected flies used in these experiments, the data do raise questions about the relative efficacy of the two vaccines, especially as only one time point was analyzed in hamsters immunized using the GLP-grade parasites, and the infections may be going up, not down at later time points.

Answer: *There is indeed comparable protection between laboratory grown LmCen^{-/-} parasites and GLP grade LmCen^{-/-} parasites in experiments where controlled infection is delivered through needle challenge. (Supplementary Fig. 8 A-B). lines 327-333.*

However, as the reviewer rightly pointed out, there can be differences in the level of protection when the infectious inoculum does differ from one batch of fly infection to others. We have provided sand fly infection status on the same day we used for transmission to test the GLP grade vaccine efficacy and this data is now incorporated in the revised manuscript (lines 338-340) and supplementary Fig. 9A-B).

However it is important to point out that even in the presence of higher parasite load in the GLP grade LmCen^{-/-} immunized hamsters after challenge (Fig.5 e-f) compared to non-GLP grade immunization and challenge experiment (Fig. 4 e-f at comparable time point of 9 months), the GLP grade LmCen^{-/-} parasites provide similar protection i.e. no mortality and lack of splenomegaly (New Fig. 5 g and d respectively) as the non-GLP grade LmCen^{-/-} (Fig.4 b-d), suggesting that efficacy of the GLP grade LmCen^{-/-} is similar to non-GLP product.

These studies also suggest that the variable parasite burden upon challenge may represent a threshold that is tolerated in the hamster model as revealed by viability and lack of splenomegaly.

Further, if there were differences in the efficacy of GLP- and non-GLP LmCen^{-/-} immunizations, we would have observed the differences early on by 6 months because by that time, in a non-immunized or partially immunized hamster model, one would have observed the lack of efficacy i.e. mortality and splenomegaly. Lack of splenomegaly would also be a practical clinical end point in future clinical trials where such variability in parasite burden is anticipated.

Fig. 5

4. Fig 4. The data sets referred to in the figure legend do not match the figure.

Answer: We regret the error and have now corrected the data set numbering, New Figure 5 g.

5. The primary response of the PBMCs from normal donors to stimulation using the LmCen-/- parasites is informative but the interpretation of the data and the conclusion that the parasites are inducing a predominantly Th1 response with suppression of IL-10 is overstated. The relative concentrations of the two cytokines measured in the culture supernatants do not necessarily reflect the relative biologic activities of the cytokines – for example a lower concentration of IL-10 may be sufficient to initiate signaling in target cells compared to IFN γ . Since levels of IL-10 were in fact detected, there is no evidence that the IL-10 response is

suppressed – compared to what? In addition, it is possible that stimulation of naïve human PBMCs in vitro with any Leishmania parasites, including *L. donovani* promastigotes, might induce the same primary response profile, so it is difficult to interpret what the signals mean in the context of vaccination.

Answer: *We agree with the reviewer’s comment about the relative concentrations of IL-10 do not reflect the biological activity. We have modified the text accordingly. Line 365-367;*

“GLP-grade *LmCen*^{-/-} parasites induced both IFN- γ (Fig. 7b) and IL-10 (Fig. 7c) in PBMCs of healthy individuals compared to their respective uninfected controls.”

Line 477-481.

“Moreover, GLP-grade *LmCen*^{-/-} parasites induced both IFN- γ and IL-10 in PBMCs of healthy human subjects living in non-endemic regions. However, the higher IFN- γ / IL-10 indicative of priming towards a Th1-biased immune response, a pre-requisite for an effective *Leishmania* vaccine was observed in human PBMCs.”

However, as the reviewer is aware, it is the ratio of IFN- γ /IL-10 that has been correlated with the vaccine efficacy (Kemp et al 1999 Clin Exp Immunol; Singh et al 2012, PNTD), Line 469-471.

“Importantly, *LmCen*^{-/-} immunized animals exhibited a significantly higher ratio of IFN- γ to IL-10 as reported in cured VL patients as a potential regulatory mechanism to control visceral infections^{32,33}”.

*Hence, we have used similar standard in our analysis for predicting the potential efficacy of *LmCen*^{-/-} parasites in human PBMCs ex vivo.*

Reviewer #2 (Remarks to the Author):

In this work, Karmakar et al. have analyzed the immunoprophylactic characteristics of a live vaccine based on the cutaneotropic species *Leishmania major*, attenuated by deletion of the genes coding for Centrin. For this, two different hamster experimental models of VL have been used: the natural infection based on sand-flies and the infection using a needle. In anticipation of possible use in humans, the immunogenicity of the vaccine has been tested in vitro, using PBMC from healthy individuals.

This is a robust and well-planned work. The experiments are well done, and the number of animals employed is adequate. The greatest strength is the high degree of protection obtained in a natural model of infection that resembles the evolution of VL in humans.

There are some issues that deserve discussion, and some of conclusions should be reformulated.

Major points.

1.- Regarding biosecurity concerns.

It is not clear whether vaccination generates a persistent infection of the attenuated strain. The detection limit

of the parasite load studies performed by limiting dilution should be indicated to avoid the apparent incongruity of saying in the results that there are no viable parasites at the site of infection (day 49 after vaccination; lines 147-152) and in the discussion that the infection is persistent (line 310-312) because of the presence of parasite DNA.

Answer: *In the revised New Fig. 1 (Please see above the response for Reviewer 1, comment:1) we have now demonstrated that vaccination does generate persistence infection of the vaccine parasite. Line 145-158,*

“In contrast, hamsters injected with *LmWT* parasites, developed ear lesions within 15 days of parasite injection that progressively increased in size (**Fig. 1a, b**). At three days post-injection, the parasite load in the ears and dLNs was similar in both *LmWT* and *LmCen*^{-/-} injected hamsters (**Fig. 1 c, d**). At 15 days post-inoculation, we started to observe a significant difference in parasite load between these groups with *LmWT* injected-hamsters displaying a higher parasite load compared to *LmCen*^{-/-}-injected hamsters (**Fig. 1 c, d**). The difference in parasite load between these two groups was greater at days 28 and 49 with parasite numbers progressively increasing in *LmWT* injected hamsters (**Fig. 1c, d**). Importantly, two of six *LmCen*^{-/-} injected hamsters cleared the parasites by 28 days post injection from the ear. Furthermore, at day 49, no viable parasites were recovered from the ears or dLNs of hamsters injected with *LmCen*^{-/-} parasites (**Fig. 1c, d**). However, when we measured the parasite burden by qPCR, low level of parasite DNA was observed both in the ear and dLN at 49-day post-*LmCen*^{-/-} injection (Supplementary Fig. 1A-C). No viable parasites were recovered from the serial dilution of spleens and livers of either *LmCen*^{-/-} or *LmWT* injected hamsters at the time points tested.”

*We showed after immune suppression with dexamethasone (New Fig. 1 h- l) some of the vaccinated animals were positive for *LmCen*^{-/-} parasites (New Fig. 1 J-L), suggesting *LmCen*^{-/-} parasites do persist in the hosts 15 weeks post immunization. Line 173-191*

“To assess the safety characteristics of *LmCen*^{-/-} parasites, we investigated lesion development and survival of *LmCen*^{-/-} parasites in immune-suppressed animals treated with dexamethasone (DXM) (**Fig. 1h**). Immune-suppressed hamsters previously injected with *LmCen*^{-/-} parasites showed no lesions at the inoculation site (ear) compared to the ulcerative lesions that developed in *LmWT* injected hamsters four weeks after immune-suppression (**Fig. 1 i, j**). Only three of the 12 *LmCen*^{-/-} injected/immune-suppressed hamsters had parasites in the inoculated ear (**Fig. 1k**) and in the dLN (**Fig. 1l**). As expected, all the *LmWT*-injected hamsters had significantly higher parasite loads in the ear (**Fig. 1k**) and in the dLN (**Fig. 1l**) compared to *LmCen*^{-/-}-injected animals (\pm immune-suppressed). To investigate whether *LmCen*^{-/-} parasites isolated from immunosuppressed animals reverted to the wild type genotype, we performed PCR analysis of the genomic DNA. We confirmed the absence of the *centrin* gene in the three isolates of *LmCen*^{-/-} parasites recovered from immune-suppressed hamsters (Supplementary **Fig. 2 A**, lanes , 3, 4 and 5, red arrow, similar to the original *LmCen*^{-/-} parasites before immunization, lane 2 red arrow). We further examined whether the *LmCen*^{-/-} parasites recovered from immune-suppressed hamsters had regained virulence by testing them in human monocyte-derived macrophages (hMDM). After 144h infection, *LmCen*^{-/-} parasites were mostly cleared from the hMDM whereas the *LmWT* parasites grew in hMDM (>6 parasites/hMDM Supplementary **Fig. 2B, C**). Collectively, these results demonstrate that the attenuated *LmCen*^{-/-} parasites are safe, unable to revert to the wild type form even in an immune-suppressed condition.”

In the revised manuscript, we have indicated that parasites were measured by serial dilution. Line 144

2. Does the infective challenge produce any effect on the parasite load at the vaccination site? In my opinion, all euthanized animals should be tested for the presence of vaccine parasites in different locations.

Answer: *In the vaccinated and challenged study (new Fig. 3), we evaluated parasite burden on the vaccinated ears at 6 months post challenge and did not find any detectable parasites using serial dilution. However, we do detect parasites in spleen and liver after challenge with wild type *L. donovani* parasites, but it is difficult to distinguish the vaccine parasites from the virulent parasites after challenge since first, the vaccine parasites are too few in number, and secondly in a potentially mixed culture of *L. donovani* and *LmCen*^{-/-} parasites it would be challenging to distinguish the two strains.*

3. Please note that concomitant immunity due to parasite persistence has been associated to the long-term immunity generated by leishmanization.

Answer: *When we immune-suppressed the immunized hamsters using dexamethasone (New Fig.1 h-l), we observed persistence of the *LmCen*^{-/-} parasites in some of the hamsters, suggesting the persistence of the vaccine parasites that could provide long range immunity against virulent challenge observed in our experiments (Figs. 3 and 4), a phenomenon similar to leishmanization.*

4. Another question. Can the presence of attenuated parasites in internal organs (liver, spleen, BM) be completely ruled out? DOI: 10.1016/j.actatropica.2005.09.007

Answer: *At different periods of post-immunization, we determined the parasite burden in spleen/ liver/lymph node and skin by serial dilution method. We did not recover any live parasites in the spleen or liver at any point post-immunization (Line 157-158) in the revised manuscript.*

“No viable parasites were recovered from the serial dilution of spleens and livers of either *LmCen*^{-/-} or *LmWT* injected hamsters at the time points tested.”

*However, we did recover *LmCen*^{-/-} parasite only in vaccinated ear and ear draining lymph nodes up to 28 days by serial dilution (New Fig.1 c-d) and by PCR at 49 days after immunization (Supplemental Fig. 1). Our data therefore suggests that there could be low persistence of *LmCen*^{-/-} parasites in other organs.*

5. Finally, If the infection is persistent, and thinking about a possible clinical trial in humans, shouldn't the effect of a possible immunosuppression on the parasite load of the vaccine parasites be studied? These considerations should be resolved experimentally or if not possible at least be discussed in the text as limitations of the study.

Answer: *In response to reviewer's comment (please see comment: 1), we demonstrated, when the immunized animals were immunosuppressed using dexamethasone there was persistence of significant number of *LmCen*^{-/-} parasites in some of the animals even after 15 weeks of immunization (New Fig. 1 k, l) indicating that the*

persistent parasites may be sequestered in as yet unknown tissues or organs. We have now revised the text accordingly lines 182-191 The persistent parasites were further characterized, and data showed that they still lacked centrin gene and had not regained virulence (New Supplementary Fig. 2 A-C). However most importantly the animals did not develop any lesion thus demonstrating the non-pathogenic nature of LmCen^{-/-} parasites (New Fig.1 I). Line 173-191

6.- Regarding the parasite dependent cytokine response.

It would be interesting to know the production of IFN-gamma and IL-10 using splenocytes and the cells of the lymph nodes draining to the vaccinated ear after stimulation with Leishmania proteins (SLA) in the vaccinated hamsters.

Answer: *We measured cytokine response in the spleen after stimulation with SLA of the immunized hamsters at 7weeks of post-immunization (New Fig.2a-b and New supplementary Fig. 3 revised text 193-207).*

“LmCen^{-/-} immunization induces protective immune response in preclinical hamster model: Towards the analysis of systemic immune responses associated with the divergent phenotype observed in *LmWT* and *LmCen^{-/-}* infections characterized by the progressive non-healing lesions and absence of lesions respectively, we analyzed the expression of several immune markers following stimulation with freeze-thaw *Leishmania* antigen (FTAg) of splenocytes. The expression profile of transcripts for pro-inflammatory Th1 type cytokines (IFN- γ , TNF- α , IL-1 β , IL-12p40 and IL-6) transcription factors (T-bet and STAT1) and chemokines/their ligands (CXCL9) was significantly higher in the *LmCen^{-/-}* immunized animals compared to *LmWT* injected hamsters (**Fig. 2 a, b; Supplementary Fig. 3**). However, splenocytes from *LmWT* infected hamsters had a significantly higher expression of both anti-inflammatory (IL-4, IL-21, GATA3 and STAT6) as well as regulatory (IL-10 and Foxp3) transcripts compared to *LmCen^{-/-}* immunized group (**Fig. 2 a, b; Supplementary Fig. 3**). The ratio of IFN- γ to IL-10 was significantly higher in both the spleen of the *LmCen^{-/-}* immunized group compared to *LmWT* injected animals (**Fig. 2b**). These results collectively suggest that *LmCen^{-/-}* immunization induces a pro-inflammatory environment in the spleen of immunized animals.”

7. Information about the SLA stimulation of spleen cells from challenged hamster (lines 219-229) are missed in the M/M section. Please, include these methods and indicate in the Fig.3 legend that cells were in vitro stimulated with SLA.

Answer: *We regret the oversight; we have now provided this information in both the revised manuscript M/M section lines 593-595 and in the figure legends of New Fig.2 and New supplementary Fig. 3.,*

“Cells were stimulated with or without *L. major* or *L. donovani* Freeze thawed antigen (FTAg), and total RNA was extracted using PureLink RNA Mini kit (Ambion) 16hours of post-stimulation.”

8.- Regarding the downregulation of the Th2 response.

The only evidence for the control of Th2 responses refers to the generic response pattern after immunization with the attenuated line in hamsters. In the rest of the experiments, the Th2 response has not been studied. Accordingly, the text must be corrected, especially in the abstract.

Answer: In the revised manuscript, we have determined the Th2 immune response (same set of cytokines measured in animals following immunization) after challenge with the wild type *L. donovani* parasites. We observed in immunized animals after six weeks of post-challenge there was reduced expression of Th2 cytokines compared to non-immunized animals. We have now incorporated this data in the revised manuscript (new Fig.3, revised text 237-266).

***LmCen*^{-/-}-immunized hamsters induce a pro-inflammatory/ Th1 type of immune response upon challenge with wild type *L. donovani*:**

Next, we wanted to investigate the efficacy of immunization with *LmCen*^{-/-} parasites against visceral leishmaniasis induced by intradermal injection of *L. donovani* parasites in hamsters (**Fig. 3 a**). The hamsters were immunized with *LmCen*^{-/-} parasites and after 7 weeks post immunization the animals were needle challenged with *LdWT* parasites and monitored for different periods (**Fig. 3 a**). Analysis of the parasite load revealed significant control of parasite numbers in the spleen (**Fig. 3 b**) and liver (**Fig. 3 c**) of *LmCen*^{-/-} immunized hamsters compared to non-immunized-infected animals at all timepoints tested. Immunized hamsters showed ~1.5 log-fold reduced parasite burden in spleen and liver as early as 1.5-month post challenge. By 12 months post-challenge, the parasite burden was reduced by ~5 log-fold for the spleen (**Fig. 3 b**) and ~12 log-fold for the liver (**Fig. 3 c**). Of note, at 12 months post challenge, 36% (4 of 11) of the spleens and 90% (10 of 11) of the livers from immunized animals had undetectable numbers of viable parasites. Taken together, these data demonstrate that *LmCen*^{-/-} elicits protection against needle infection in a hamster model of VL.

To characterize the immune correlates of protection, we measured the gene expression profile *ex-vivo* after antigen re-stimulation for the spleen by qPCR following 1.5 months after needle challenge with virulent *L. donovani* parasites. Evaluation of the immune response in the spleen (same genes that were tested before challenge) also showed a markedly increased expression of pro-inflammatory cytokine, chemokine and transcriptional factor transcripts (IFN- γ , TNF- α , IL-12p40, T-bet, STAT1 and CXCR3) with a significant decrease in anti-inflammatory genes (IL-21 and STAT6) in the immunized-challenged group compared to the non-immunized challenged group (**Fig. 3d, e**; Supplementary **Fig. 6**). Of Note, expression of M2 macrophage phenotype genes (CCL17) and chemokine and chemokine receptor transcripts (CCR4 and CCL4) were higher in the non-immunized challenged group compared to non-immunized challenged group (**Fig. 3 d**, Supplementary **Fig. 6**). Importantly, the ratio of IFN- γ to IL-10 was significantly higher in the spleen of the immunized-challenged group compared to non-immunized challenged animals (**Fig. 3 e**). Collectively, these results indicate generation of a proinflammatory type of immune response following needle challenge in the *LmCen*^{-/-} immunized animals that confers protection against virulent *L. donovani*.

Fig. 3

9. Respectfully, I do not agree with the interpretation of the results of the PBMC in vitro stimulation. The results show that after infection with the vaccine parasites, both IFN-gamma (F6b) and IL-10 (F6c) are produced. In other words, the vaccine generates the secretion of both cytokines. The next parts of the text must be corrected:

Results (277-280). GLP-grade *LmCen*^{-/-} parasites induced higher (not lower) levels of IL-10 (Fig. 6c) in PBMCs of healthy individuals compared to their respective uninfected controls,
Discussion (369) GLP *LmCen*^{-/-} do not induce IL-10 suppression in PBMCs, just the opposite occurs.

Answer: *We agree with reviewer's comment and have modified the text accordingly in the revised manuscript Line 365-367,*

“GLP-grade *LmCen*^{-/-} parasites induced both IFN- γ (Fig. 7b) and IL-10 (Fig. 7c) in PBMCs of healthy individuals compared to their respective uninfected controls.”

Line 477-481.

“Moreover, GLP-grade *LmCen*^{-/-} parasites induced both IFN- γ and IL-10 in PBMCs of healthy human subjects living in non-endemic regions. However, the higher IFN- γ / IL-10 indicative of priming towards a Th1-biased immune response, a pre-requisite for an effective *Leishmania* vaccine was observed in human PBMCs.”

10. Consistent with the authors' conclusions, I agree that the production of IFN-gamma is greater than that of IL-10, but both cytokines are produced after parasite-driven stimulation

Answer: *We thank the reviewer for agreeing with our conclusions.*

11. The confusion is even greater in the text of the abstract where it is stated that: "Downregulation of Th2 response in the PBMCs, isolated from healthy people from non-endemic region, upon *LmCen*^{-/-} infection was also observed".

Answer: *We agree with reviewer's comment and modified the text in the abstract accordingly in the revised manuscript. Line 58-59.*

“PBMCs, isolated from healthy people from non-endemic region, upon *LmCen*^{-/-} infection also induced more IFN- γ compared to IL-10 further confirms the potency of this vaccine in human use.”

12. In addition to the already reported inaccurate interpretation of the results, the authors assign IL-10 the ability to define a Th2 response. Please note that this cytokine is produced by other cell types besides Th2 lymphocytes, even activated Th1 lymphocytes! Please edit the abstract to reflect the results more accurately.

Answer: *We agree with reviewer's comment and have modified the text accordingly in the revised abstract. Line 54-59.*

“Spleen cells from *LmCen*^{-/-} immunized and *L. donovani* challenged hamsters produced significantly higher Th1-associated cytokines including IFN- γ , TNF- α , and reduced expression of the anti-inflammatory cytokines like IL-10, IL-21, compared to non-immunized challenged animals. PBMCs, isolated from healthy people

from non-endemic region, upon *LmCen*^{-/-} infection also induced more IFN- γ compared to IL-10 further confirms the potency of this vaccine in human use.”

13. To implicate down-regulation of the Th2 mediated responses in the induced protection in hamster or in PBMC assays it is necessary to include data regarding the expression of Th2 cytokines in the stimulation experiments discussed above (section 2) or in this section.

Answer: *We have measured Th2 cytokine responses in immunized hamsters at 7weeks of post-immunization (New Fig.2a-b and New supplementary Fig. 3, and revised text 193-207). We also have measured IL-10 levels in normal human PBMCs. More Th2 cytokines in human PBMCs will be studied in the future.*

14.- Anti leishmania humoral responses.

In addition to my comments of section 3, it should be also considered that in the hamster model, as in humans affected by VL, the decrease in the antibody titer (IgG) against parasite proteins is correlated with the evolution of less severe forms of the disease as well as with vaccine-induced protection. Thus, it would be interesting to know the anti-leishmania antibody response elicited in the different groups of animals used in this work, or at least to know why the authors have not included these data.

Answer: *We agree with the reviewer’s comment and have now incorporated the anti-leishmanial antibody titer in *LmCen*^{-/-} immunized animals and compared to wild type *L. major* parasite infected animals at seven weeks either post immunization or post infection. We observed high titer of IgG2a (indicative of Th1) in the immunized animals while high titer of IgG1 (indicative of Th2) in the wild type parasite infected animals. We incorporated this information in the revised manuscript (Lines 227-235) and added the data in the New supplementary Fig. 5).*

“Furthermore, from the analysis of immunoglobulin subtypes one can predict the outcome of the immune response, a IgG1 dominant response predictive of Th2 and a IgG2a dominant response predictive for Th1 type of immune response³⁰. We measured anti-leishmanial IgG1 and IgG2a from the serum of *LmCen*^{-/-} or *LmWT* parasite infected hamsters at 49days of post-infection (Supplementary **Fig. 5 A, B**). The level of IgG1 was significantly higher in *LmWT* injected hamsters compared to *LmCen*^{-/-} immunized hamsters (Supplementary **Fig. 5 A**). In contrast IgG2a was significantly higher in *LmCen*^{-/-} immunized hamsters compared to *LmWT* infected hamsters (Supplementary **Fig. 5 B**) resulting in higher ratio of IgG2a/ IgG1 in *LmCen*^{-/-} immunized hamsters (Supplementary **Fig. 5 C**).”

Supplementary Figure 5

Supplementary Figure 5. *LmCen*^{-/-} immunization induces IgG2a type of immune response.

A, B, Anti-leishmanial IgG1 (A) and IgG2a (B) were measured from the serum of hamsters immunized with *LmCen*^{-/-} parasites (n=8) and compared with *LmWT* (n=9) infected group following 7 weeks of post inoculation. (C) The ratio of IgG2a/IgG1. Results (mean ± SD) are representative of cumulative effect of two independent experiments. Statistical analysis was performed by non-parametric Mann-Whitney two-tailed test.

Minor

1. The quality of figure 2 should be improved.

Answer: We have reconfigured Figure 2 a-c and supplementary Fig. 3 and supplementary Fig. 4 A-C showing data underlying the analysis. The text is modified accordingly (Lines 193-207).

2. Consider editing lines 81-82 for more clarity: In contrast, disease progression is associated with a dominant Th2 type as well as IL-10 mediated responses.

Answer: We modified this line in the revised manuscript. Line 82-83,

“ In contrast, disease progression is associated with a dominant Th2 type as well as IL-10 mediated response⁷”.

Reviewers' Comments:

Reviewer #1:

Remarks to the Author:

Most of my substantive comments, in particular those pertaining to the key data generated in hamsters, have been adequately addressed in the revised manuscript and in the author responses. I still feel that they continue to over interpret the results involving the PBMCs from normal donors that are stimulated with infected autologous adherent cells. Their conclusion that the IFN γ / IL-10 ratios of the concentration of secreted cytokines confirm the potency of the vaccine is overstated. Absent data regarding the relative biological activities of these cytokines on their respective target cells, the ratio of the cytokine concentrations in the supernatants is not especially meaningful. There are a number of mouse models in which the frequency IFN γ cells far exceeds the number of IL-10 producing cells, yet the mice will fail to heal unless IL-10 is neutralized. Furthermore, in these PBMC assays, it is possible that any Leishmania strain, including wild type *L. donovani* strains, would elicit a similar response profile.

Reviewer #2:

Remarks to the Author:

This reviewer appreciates the great effort made by the authors to edit and improve their manuscript. All the questions raised have been adequately answered and the manuscript in its current version has reached a high level of quality and impact. For these reasons, I consider that the work can be published in its current form.

REVIEWERS' COMMENTS:

Reviewer #1 (Remarks to the Author):

Comment: Most of my substantive comments, in particular those pertaining to the key data generated in hamsters, have been adequately addressed in the revised manuscript and in the author responses. I still feel that they continue to over interpret the results involving the PBMCs from normal donors that are stimulated with infected autologous adherent cells. Their conclusion that the IFN γ / IL-10 ratios of the concentration of secreted cytokines confirm the potency of the vaccine is overstated. Absent data regarding the relative biological activities of these cytokines on their respective target cells, the ratio of the cytokine concentrations in the supernatants is not especially meaningful. There are a number of mouse models in which the frequency IFN γ cells far exceeds the number of IL-10 producing cells, yet the mice will fail to heal unless IL-10 is neutralized. Furthermore, in these PBMC assays, it is possible that any Leishmania strain, including wild type *L. donovani* strains, would elicit a similar response profile.

Response to the Comment: *We agree with the reviewer's skepticism on the validity of IFN- γ /IL-10 ratio as a rigorous standard of vaccine potency especially from the in vitro human PBMC studies as performed in this manuscript. The full spectrum of cytokine responses, the cellular sources and their coordinated activities following immunization will be investigated in future clinical trials towards discovering biomarkers of vaccine potency and efficacy.*

*However, the reviewer's assertion that levels of IFN- γ observed in our studies may not be adequate to neutralize the effects of IL-10 does not apply in a strict sense since a higher ratio of IFN- γ to IL-10 in the *LmCen*^{-/-} infected human PBMCs compared to the non-infected human PBMCs is consistent with our hamster studies analyzing the immunogenicity before challenge (Figure 2) and further showed strong protection upon virulent challenge with *L. donovani* (Figure 3 and 4).*

Accordingly, we have revised the text to highlight that our human PBMC data shows concordance with immunogenicity studies in preclinical models. Lines: 57-61, 348-352, 458-462 and 466-468.

Reviewer #2 (Remarks to the Author):

Comment: This reviewer appreciates the great effort made by the authors to edit and improve their manuscript. All the questions raised have been adequately answered and the manuscript in its current version has reached a high level of quality and impact. For these reasons, I consider that the work can be published in its current form.

Response: We thank reviewer for appreciation our work and effort.